# Tumor invasion in draining lymph nodes is associated with Treg accumulation in breast cancer patients

Nicolas Gonzalo Núñez [1,7,9], Jimena Tosello Boari [1,9], Rodrigo Nalio Ramos [1], Wilfrid Richer[1], Nicolas Cagnard[2], Cyrill Dimitri Anderfuhren [3], Leticia Laura Niborski[1], Jeremy Bigot[1], Didier Meseure[4,5], Philippe De La Rochere[1], Maud Milder[4,5], Sophie Viel[1], Delphine Loirat[1,5,6], Louis Pérol[1], Anne Vincent-Salomon[4,5], Xavier Sastre-Garau[4,8], Becher Burkhard [3], Christine Sedlik [1,5], Olivier Lantz [1,4,5], Sebastian Amigorena[1,5] & Eliane Piaggio [1,5✉]

Tumor-draining lymph node (TDLN) invasion by metastatic cells in breast cancer correlates with poor prognosis and is associated with local immunosuppression, which can be partly mediated by regulatory T cells (Tregs). Here, we study Tregs from matched tumor-invaded and non-invaded TDLNs, and breast tumors. We observe that Treg frequencies increase with nodal invasion, and that Tregs express higher levels of co-inhibitory/stimulatory receptors than effector cells. Also, while Tregs show conserved suppressive function in TDLN and tumor, conventional T cells (Tconvs) in TDLNs proliferate and produce Th1-inflammatory cytokines, but are dysfunctional in the tumor. We describe a common transcriptomic signature shared by Tregs from tumors and nodes, including CD80, which is significantly associated with poor patient survival. TCR RNA-sequencing analysis indicates trafficking between TDLNs and tumors and ongoing Tconv/Treg conversion. Overall, TDLN Tregs are functional and express a distinct pattern of druggable co-receptors, highlighting their potential as targets for cancer immunotherapy.

[1] Institut Curie, PSL Research University, INSERM U932, F-75005 Paris, France. [2] Paris-Descartes Bioinformatics Platform, 75015 Paris, France. [3] Institute of Experimental Immunology, University of Zurich, Winterthurerstr. 190, CH-8057 Zurich, Switzerland. [4] Institut Curie, PSL Research University, Departement de Biologie des Tumeurs, F-75005 Paris, France. [5] Centre d'Investigation Clinique Biotherapie CICBT 1428, Institut Curie, Paris F-75005, France. [6] Institut Curie, PSL Research University, Departement d'Oncologie Medicale, F-75005 Paris, France. [7] Present address: Institute of Experimental Immunology, University of Zurich, Winterthurerstr. 190, CH-8057 Zurich, Switzerland. [8] Present address: Institut de Cancerologie de Lorraine Department of Biopathology, 6, avenue de Bourgogne CS 30519, 54519 Vandoeuvre-lès-Nancy cedex, France. [9] These authors contributed equally: Nicolas Gonzalo Núñez, Jimena Tosello. ✉email: eliane.piaggio@curie.fr

In human breast cancer, regional LNs are frequently the first site of metastasis. From a clinical standpoint, the tumor invasion of tumor-draining LNs (TDLNs) is an important step in disease progression, and is a prognostic indicator of the risk of recurrence and of poor survival[1–3]. However, recent clinical trials in breast cancer indicate that LN resection does not increase the patients' overall survival[4,5]. From an immunologic standpoint, little is known about the consequences of tumor metastasis on the LN's immune functions. However, it is clear that tumors develop a broad array of immunosuppressant mechanisms. This fact raises the question of whether the invasion of TDLNs during tumor spreading produces an accumulation of immunosuppressive cells—like FOXP3+ CD4+ regulatory T cells (Tregs)—and imparts a tolerogenic micro-environment[1,2,6,7]. At present, most of our knowledge of the immune status of breast cancer patients comes from the analysis of primary tumor and blood samples, whereas data on the immune characteristics of TDLNs are scarce[8–11].

Tregs expressing the transcription factor Foxp3 maintain self-tolerance and homeostasis of immune system but also limit sterilizing immunity and dampen antitumor immunity[12,13]. In breast cancer increased numbers of tumor-infiltrating Tregs correlate with reduced survival[12,14,15]. The balance between Tregs and Tconvs largely defines the outcome of the immune response. Tregs are highly heterogenous; they can arise in the thymus or can emerge in the periphery from Tconvs. Furthermore, upon activation, Tregs can acquire different phenotypes, associated to different functions. Thus, based on the expression of some chemokine receptors and transcription factors, Tregs, similar to T helper cells (Th) can be further classified in Tr1, Tr2, Tr17, Tfr[13,16,17]. Only a few studies have analyzed the phenotype[8,12] and function of the Tregs present in the TDLNs of patients with breast cancer[18], and the available data suggest that high frequencies of total CD4+ T cells and low frequencies of Tregs[18–20] are associated with a good prognosis. However, a comprehensive global analysis of the phenotype and function of Tregs in tumor-invaded (I) TDLNs, non-invaded (NI) TDLNs, and primary luminal breast cancer tumors (T) is still missing.

The immunotherapeutic blockade of immune checkpoints, such as PD-1/PD-L1 and CTLA-4, has given impressive clinical results and manageable safety profiles in various tumor types[21, 22]. In patients with breast cancer, one of the most encouraging immunotherapies is anti-PD-1/PD-L1 monotherapy; it achieves an objective response rate of between 12 and 21%[23]. However, a CTLA-4-blocking antibody has shown only limited clinical benefit[24]. The expression pattern of immune checkpoints of Tregs and Tconvs in luminal breast cancer has been poorly studied[12,25]. Thus, considering that checkpoint-blocking antibodies can act not only during the effector T-cell phase in the tumor bed[26], but also during T-cell priming in the TDLNs, it is important to understand the immune status of the T cells (including Tregs), that can also be targeted by these antibodies.

In the present study, we use high-dimensional flow cytometry, functional assays, T-cell receptor (TCR) repertoire analysis, and RNA sequencing to characterize the immune phenotype, function and dynamics of Tregs present in paired I and NI TDLNs, and tumor; and identified CD80-expressing Tregs as a subset of Tregs associated with bad prognosis in breast cancer patients. These data bring insights into the immunomodulatory mechanisms associated to the presence of the tumor, and should be instrumental to guide the rationalized design of improved immunotherapies.

## Results

### "Bona fide" memory Treg cells accumulate in metastatic TDLNs.
We characterized and compared the immune profile of Tregs from paired NI and I TDLNs from patients with breast cancer. As reference, we also analyzed the Tconvs present in the primary tumor, which have already been partly characterized[12,25,27]. To this end, we immune profiled freshly resected NI and I TDLNs and primary tumors from patients with luminal breast cancer having undergone standard-of-care surgical resection (Fig. 1a). The patients' clinical and pathological data are summarized in Supplementary Table 1.

TDLNs were classified as NI or I, based on the absence or presence of metastatic tumor cells (identified as EPCAM+ and CD45−). The flow cytometry results were confirmed by pathologic assessment (Fig. 1b, c). We evaluated the overall distribution of the different immune cell populations in the three tissues. Using an unsupervised data analysis (see methods), we identified eight main clusters, including B cells, Tconvs, Tregs, CD8+ T cells, no CD4+ CD8+ T cells, fibroblast/endothelial cells, tumor cells, and other leukocytes (Fig. 1d, e and Supplementary Fig. 1b). All clusters were subsequently confirmed by manual gating (Supplementary Fig. 1b–c). Quantification of the different populations (Supplementary Fig. 1d–f and Supplementary Table 2) indicated that the only population that significantly changed between NI and I TDLNs was the Tregs (Fig. 1f). Although there is some heterogeneity within samples, for 10 out of 14 patients the proportion of Tregs among total CD45+ T cells was higher in the I TDLN, compared with the NI one ($p < 0.05$). Also, we observed that in the primary tumor, Tregs constitute an important proportion of the whole T-cell infiltrate, reaching up to 19% of CD45+ cells.

We next analyzed the impact of nodal metastasis on the naïve/memory phenotype of Tregs in the TDLNs. As reported by Sakaguchi and colleagues[28,29], Tregs and Tconvs can be classified based on the expression of CD45RA and FOXP3 as: (I) naïve Tregs, (II) effector Tregs (Eff Treg), (III) FOXP3+ non-Tregs (recently activated Tconvs), (IV) memory Tconvs, and (V) naïve Tconvs (Fig. 2a). We observed that I TDLNs contained higher frequencies of Eff Tregs ($p < 0.01$), and lower proportions of naïve Tconvs ($p < 0.05$) than NI TDLNs, for most of the patients studied. In the tumor, the CD4+ T-cell compartment was characterized by very low proportions of naïve cells and relatively high proportions of Eff Tregs and memory Tconvs[12,30]. These results let us hypothesize that tumor cells in the TDLNs (i) may be recognized by naïve Tregs/Tconvs, which then acquire an Eff Treg phenotype; and/or (ii) may be involved in the accumulation of Eff Tregs that circulate from the tumor to the TDLNs.

### Tregs in TDLNs show distinct immune checkpoint molecules.
Little information exists on the pattern of expression of druggable immune checkpoint molecules on Tregs from TDLNs and tumor of patients with breast cancer[12,25]. Given that immune checkpoint molecules were barely detected in naïve T cells (Supplementary Fig. 2a), we performed their analysis on the memory T-cell compartment. Among the Tregs of the three tissues, a very high frequency expressed the co-inhibitory receptors PD-1 and CTLA-4 (Supplementary Fig. 2a–c) and the costimulatory receptors ICOS and GITR (Fig. 2d and Supplementary Fig. 2c); and a lower fraction expressed OX40 (Fig. Supplementary Fig. 2c). Invasion of TDLNs correlated with higher frequencies of Tregs expressing ICOS, GITR, OX40, and PD-1, increased frequencies of Tregs with double expression of PD-1 and ICOS (PD-1+ ICOS+ Tregs) (accompanied by a decrease in the PD-1−ICOS− Tregs) (Fig. 2e), and no significant difference for CTLA-4 expression (always very high) (Fig. 2b–d and Supplementary Fig. 2c). In the tumor, even higher frequencies of Tregs expressing each of the measured checkpoints were observed (Fig. 2b–d and Supplementary Fig. 2c). These results suggest that in the presence of tumor cells, higher proportion of Tregs get activated and express immune

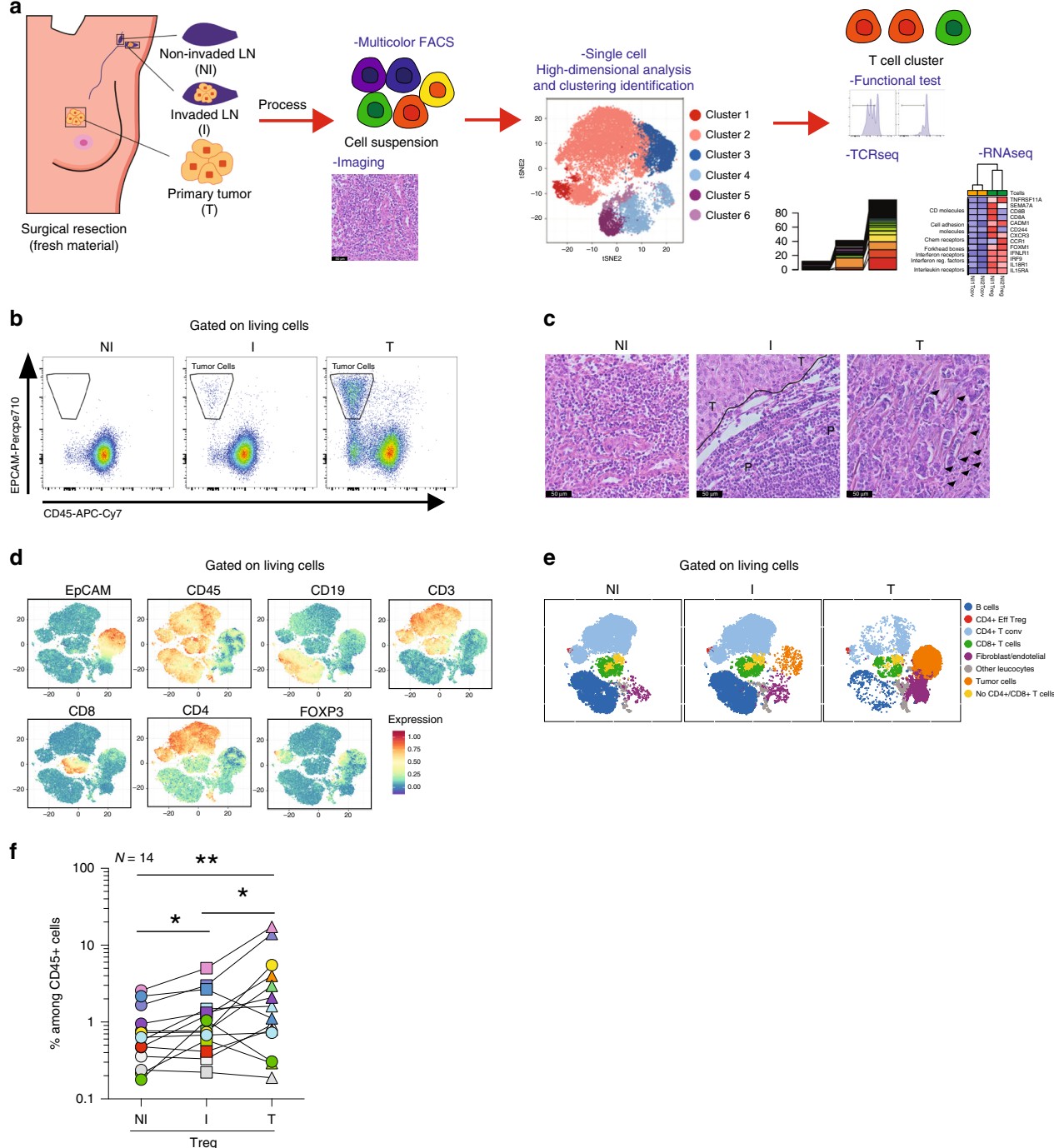

**Fig. 1 Overall distribution of immune cell populations in NI and I TDLNs, and tumor. a** Flowchart of the study design. Non-invaded (NI) and invaded (I) TDLNs, and the primary tumor (T) were collected and samples were split and processed for hematoxylin–eosin staining or for flow cytometry analysis, functional tests or transcriptome and TCR sequencing. **b** A representative flow cytometric analysis of EPCAM+ CD45+ cell populations in NI and I TDLNs and primary tumors. **c** Representative hematoxylin–eosin staining of NI TDLNs (left panel), I TDLNs (middle panel), and primary tumors (right panel) (P: peritumor area; T: tumor area; arrows: leukocytes) from one out of three independent experiments with similar results. **d** Samples were stained with EpCAM, CD45, CD19, CD3, CD8, CD4, and FOXP3 and analyzed by FACS, and shown is a t-SNE map displaying randomly selected cells from TDLNs and primary tumors. **e** t-SNE map showing the FlowSOM-guided clustering of NI, I, and T cells. Each color represents a cluster and is associated with a different immune population. **f** Frequencies of Tregs among CD45+ cells in TDLNs and tumor (NI vs I TDLNs $p = 0.0134$; NI TDLN vs T $p = 0.004$; I TDLN vs T $p = 0.0494$). Wilcoxon matched-pairs signed rank test, *$p < 0.05$, **$p < 0.01$ ($n = 14$).

checkpoints, and underscore that antibodies targeting these molecules could act not only during the effector phase in the tumor, but also during priming, in the TDLNs.

As T-cell subsets with opposed immune function can be modulated by therapeutic Abs with agonistic, antagonistic, or depleting functions, it is of interest to understand the differential expression of positive and negative immune checkpoints on effector Tconvs and CD8+ T cells, and on suppressive Tregs (Supplementary Fig. 2A and 3). At all sites, Tregs were the most prevalent T-cell subset population expressing costimulatory

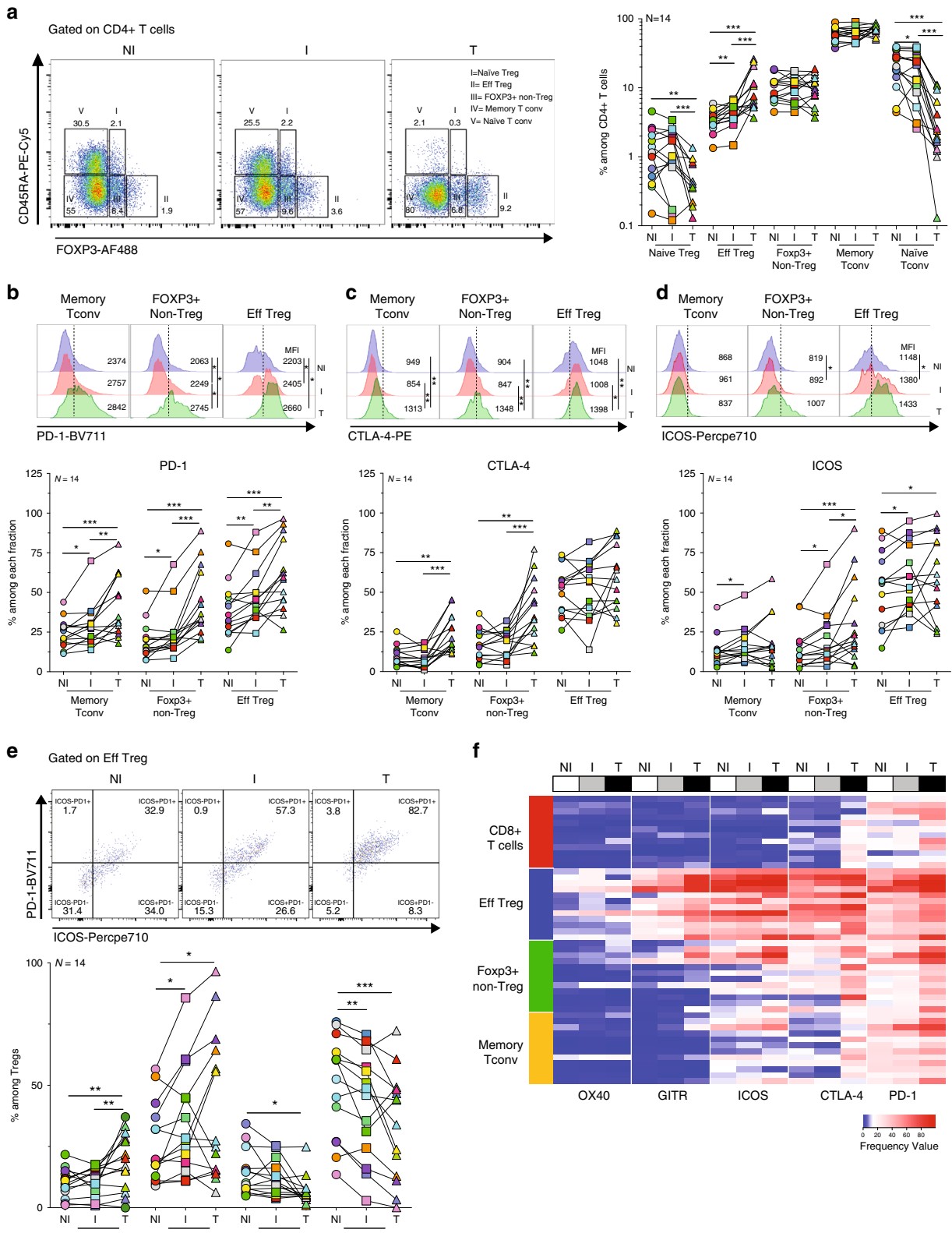

(ICOS, GITR, and OX40) and co-inhibitory (PD-1, CTLA-4) immune checkpoint molecules (Fig. 2b–e and Supplementary Fig. 2c–e). Also, Tregs showed the highest mean fluorescence intensities (MFIs) values for ICOS, OX40, GITR, CTLA-4, and PD-1 (Supplementary Fig. 2a). Tregs also expressed the highest levels other Treg cell-associated molecules, such as Ki-67,

HELIOS, CD25, and CD39 (Supplementary Fig. 2a–b), which could represent additional druggable targets. The higher CD39 expression in tumor and I LN Tregs than the NI LN counterparts (Supplementary Fig. 2b), indicates that the presence of tumor cells triggers the activation of Tregs and imprints a potential higher suppressive function.

**Fig. 2 Naive/memory phenotype and immune checkpoint expression of T cells in TDLNs, and tumor.** Cell suspensions of TDLNs cells and tumors were stained for CD3, CD4, CD8, CD45RA, FOXP3, PD-1, CTLA-4, and ICOS. **a** Representative flow cytometric analysis (left panel) and frequency (right panel) of cells distinctively expressing CD45RA and/or FOXP3 among CD4+ T cells in TDLNs and tumor (Naive Treg: NI TDLN vs T $p = 0.0023$; I TDLN vs T $p = 0.0004$; Eff Treg: NI vs I TDLNs $p = 0.0017$; NI TDLN vs T $p = 0.0004$; I TDLN vs T $p = 0.0009$; naive Tconv: NI vs I TDLNs $p = 0.0494$; NI TDLN vs T $p = 0.0002$; I TDLN vs T $p = 0.0001$). Wilcoxon matched-pairs signed rank test. **b–d** Representative histograms and frequencies of **b** PD-1 (memory Tconv: NI vs I TDLNs $p = 0.0436$; NI TDLN vs T $p = 0.0009$; I TDLN vs T $p = 0.0012$; Foxp3+ non-Treg: NI vs I TDLNs $p = 0.0203$; NI TDLN vs T $p = 0.0001$; I TDLN vs T $p = 0.0002$; Eff Treg: NI vs I TDLNs $p = 0.0031$; NI TDLN vs T $p = 0.0002$; I TDLN vs T $p = 0.0017$), **c** CTLA-4 (memory Tconv: NI TDLN vs T $p = 0.0023$; I TDLN vs T $p = 0.0004$; Foxp3+ non-Treg: NI TDLN vs T $p = 0.0031$; I TDLN vs T $p = 0.0004$) and **d** ICOS (memory Tconv: NI vs I TDLNs $p = 0.0107$; FOXP3+ non-Treg: NI vs I TDLNs $p = 0.0295$; NI TDLN vs T $p = 0.0009$; I TDLN vs T $p = 0.0245$; Eff Treg: NI vs I TDLNs $p = 0.0101$; NI TDLN vs T $p = 0.0436$) expression among the indicated CD4+ T cell subpopulations. Wilcoxon matched-pairs signed rank test. **e** Representative flow cytometric analysis (upper panel) and frequency (lower panel) of ICOS and/or PD-1 among Eff Tregs (ICOS-PD-1+: NI TDLN vs T $p = 0.0031$; I TDLN vs T $p = 0.0022$; ICOS + PD-1+: NI vs I TDLNs $p = 0.0215$; NI TDLN vs T $p = 0.0245$; ICOS + PD-1-: NI vs I TDLNs $p = 0.0134$; ICOS-PD-1-: NI vs I TDLNs $p = 0.0023$; NI TDLN vs T $p = 0.0006$). Wilcoxon matched-pairs signed rank test. **f** Heatmap displaying the frequency of expression of OX40, GITR, ICOS, CTLA-4, and PD-1 on the indicated T-cell subpopulations in TDLNs and tumor ($N = 12$). Red and blue indicate higher and lower expression frequencies, respectively. Wilcoxon matched-pairs signed rank test, $*p < 0.05$, $**p < 0.01$, $***p < 0.001$.

**Tregs in TDLN and tumor have conserved suppressive function.** Tregs can show an unstable phenotype and lose their suppressive function in pro-inflammatory microenvironments[31,32]. We evaluated whether NI, I TDLNs and tumor, represent pro- or antiinflammatory microenvironments in which Tregs exert their functions. Ex vivo whole-cell suspensions from the three tissues were stimulated and T-cell proliferation was analyzed. As shown in Fig. 3a, Tconvs from NI and I TDLNs highly proliferated; however, Tconvs from the tumor stopped dividing after a few cycles. Upon phorbol 12-myristate 13-acetate (PMA)-ionomycin stimulation, the frequency of IFN-γ+ Tconvs was significantly higher in I versus NI TDLNs (Fig. 3b), and only a small fraction of all the T cell produced IL-17A. To assess whether Tregs from I TDLN -where Tconvs are more pro-inflammatory- maintain their suppressive function, we performed classical suppression tests. Tregs from both I TDLNs and the tumor had similarly proficient suppressor functions (Fig. 3c).

Overall, these results indicate that: (i) Tregs from both TDLNs and the tumor had conserved ex vivo suppressor functions, and (ii) that although Tconvs from I TDLNs readily produce pro-inflammatory cytokines, the Tregs there do not, indicating that they did not acquire pro-inflammatory function in invaded breast cancer LNs.

**I TDLNs show higher proportions of Tr1 and Tfr cells.** Published transcriptomic signatures of Tregs from different tissues indicate that Tregs constitute a heterogeneous population shaped by microenvironmental cues[12,25,33]. Indeed, the role of cytokines and chemokines present in the microenvironment influences the terminal differentiation of tumor-specific Tconvs with different polarization patterns, functional specialization, and migration capacities. In addition, mouse studies have suggested that Tregs may also co-opt transcriptional programs adapted to regulate Th1, Th2, or Th17 responses[13,16,17].

To assess the functional diversity of Tconvs and Tregs present in both I and NI TDLNs, we analyzed CD4+ T-cell chemokine receptors pattern (as described in[16,34]). We first evaluated the phenotype of Tconvs (CD4+ CD127+ CD25−) as Th1 (CXCR3+), Th2 (CXCR3−CCR6−CCR4+), Th17 (CXCR3−CCR6+CCR4+ CCR10−), Th22 (CXCR3−CCR6+CCR4+CCR10+), and T follicular helper cells (Tfh, PD-1^high^CXCR5+) (Fig. 4a). Among Tconvs (Fig. 4b), Th1 cells were the most abundant population. Compared with NI TDLNs, I ones showed higher proportions of Th1 ($p < 0.01$) and Tfh cells ($p < 0.05$), and lower proportions of Th22 cells ($p < 0.05$). Around 10% of Tconvs were Th2 cells (with no differences between NI and I TDLNs) and very few of them expressed CRTh2, a marker of a subset of lineage-committed Th2 cells[34]. Finally, Th17 cell proportions were similar in NI and I TDLNs (Fig. 4b).

We observed that Tregs from TDLNs followed a similar chemokine receptor pattern as Tconvs. And also, similar to Tconvs, compared with NI TDLNs, in I TDLNs there were significant higher proportions of Th1-like Tregs ($p < 0.05$), Th22-like Tregs ($p < 0.05$) and Tfh-like Tregs (Tfr) ($p < 0.05$), respectively. Similar frequencies of Th2-like Tregs and Th17-like Tregs were found in NI and I TDLNs (Fig. 4c). The strikingly similar pattern of helper Tconv and helper-like Treg phenotype was reflected by a strong positive correlation of the frequencies of Th1 vs Th1-like ($r^2 = 0.23$; $p = 0.008$), Th2 vs Th2-like ($r^2 = 0.36$; $p = 0.0005$), Th22 vs Th22-like ($r^2 = 0.41$; $p = 0.007$), and Tfh vs Tfr ($r^2 = 0.92$; $p < 0.0001$) populations; but not of Th17 vs Th17-like ($r^2 = 0.08$; $p = 0.23$) (Fig. 4d). The permeabilization step required for FOXP3 staining is not compatible with labeling of chemokine receptors, except for CXCR3 and CCR4. We further studied the expression of these two receptors in the CD4+ T-cell subsets as described in Fig. 2a. We observed that the frequency of CXCR3+ cells was significantly increased among Eff Tregs (57.9% vs 67.5%, $P < 0.01$), FOXP3+ non-Tregs (38.9% vs 47%, $P < 0.01$) and memory Tconvs (42.3% vs 53.5%, $P < 0.01$) from the I compared with NI TDLNs (Supplementary Fig. 4a) and we observed similar frequencies of CCR4+ CD4+ T-cell subpopulations in I vs NI TDLNs (Supplementary Fig. 4b). We analyzed the expression of T-bet and GATA3, two canonical molecules associated with Th1-like and Th2-like functions, respectively (Supplementary Fig. 4c–d). In I TDLNs we observed the frequency of T-bet+ Tregs was higher and the frequency of GATA3+ Tregs was lower compared with NI TDLNs. These results suggest that nodal invasion by the tumor induces a Th1-biased phenotype of Treg cells, and point to a shared migrational imprinting program with effector (Th1) Tconvs, like previously described[16,34].

In all, Tconvs and Tregs from TDLNs follow similar phenotypic and functional specific programs, reinforcing the concept that the ability of Tregs to maintain local immune homeostasis depends on their appropriate colocalization with Tconvs.

**Tregs with shared TCRs are found in TDLNs and tumors.** The TCR repertoire has been used to explore the clonal diversity and trafficking patterns among TDLNs and tumor of total T cells[35]. To go deeper in CD4+ T cells subsets, we performed high-throughput sequencing of the TCR-β CDR3s of Tregs (CD25^high^CD45RA−) and Tconvs (CD25−CD45RA−) (see methods) obtained from the NI and I TDLNs for five patients and from primary tumors for three of these patients (the distribution of the sequenced TCR-β CDR3s for the entire set of samples are summarized in Supplementary Table 3). The cumulative

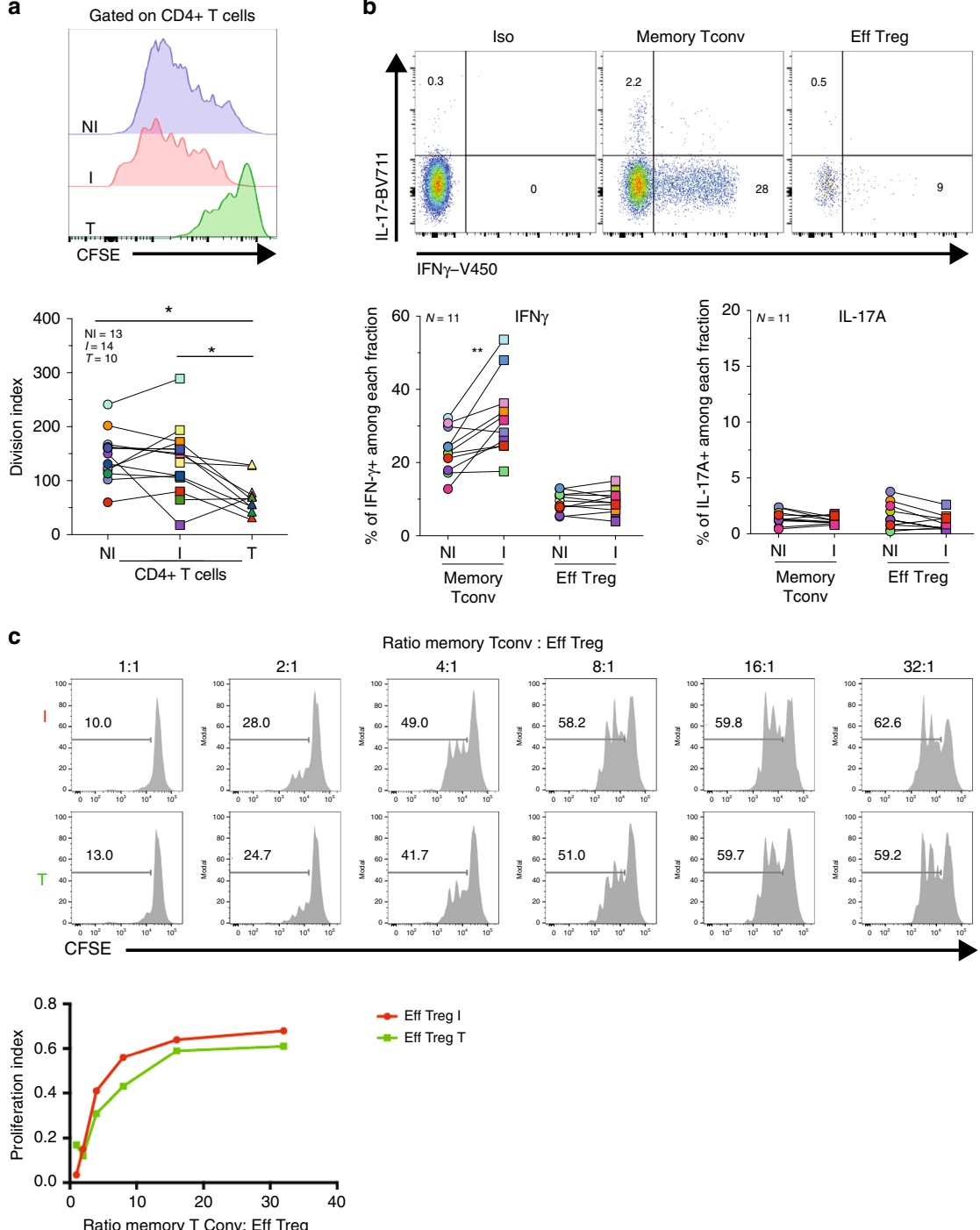

**Fig. 3 Analysis of CD4+ Tconv and Treg function in TDLNs.** Axillary TDLNs cells and tumors were stained with CFSE, ex vivo stimulated with anti-CD3/CD28 beads for 96 h, and stained with CD3, CD4, CD8, and FOXP3. **a** Representative histograms showing CFSE dilution (left panels) and the division index (right panels) of CD4+ T cells in TDLNs and tumor ($p = 0.0156$ NI TDLN vs T; $p = 0.0273$ I TDLN vs T). Wilcoxon matched-pairs signed rank test. **b** Cell suspensions of TDLNs were ex vivo stimulated with PMA/Ionomycin for 4 h and stained for CD3, CD4, IFN-g, and IL-17A. Shown is a representative flow cytometric analysis (upper panel) and frequency (lower panel) of IFN-γ and IL-17 among gated CD4+ T cell subpopulations from TDLN ($p = 0.0029$ NI vs I TDLNs). **c** Evaluation of the suppression of autologous memory Tconv (CD4+ CD25−) proliferation by Tregs (CD4+ CD25high) sorted from fresh I TDLNs (I) and the corresponding primary tumor (T). Cells were cultured for 4 days in the presence of anti-CD3/CD28 beads. Histograms show the CFSE dilution (upper panel) and proliferation index of Tconvs in the presence or absence of Tregs at the indicated ratios (lower panel). Representative histograms ($N = 1$) from one out of two independent experiments with similar results. Wilcoxon matched-pairs signed rank test, *$p < 0.05$, **$p < 0.01$.

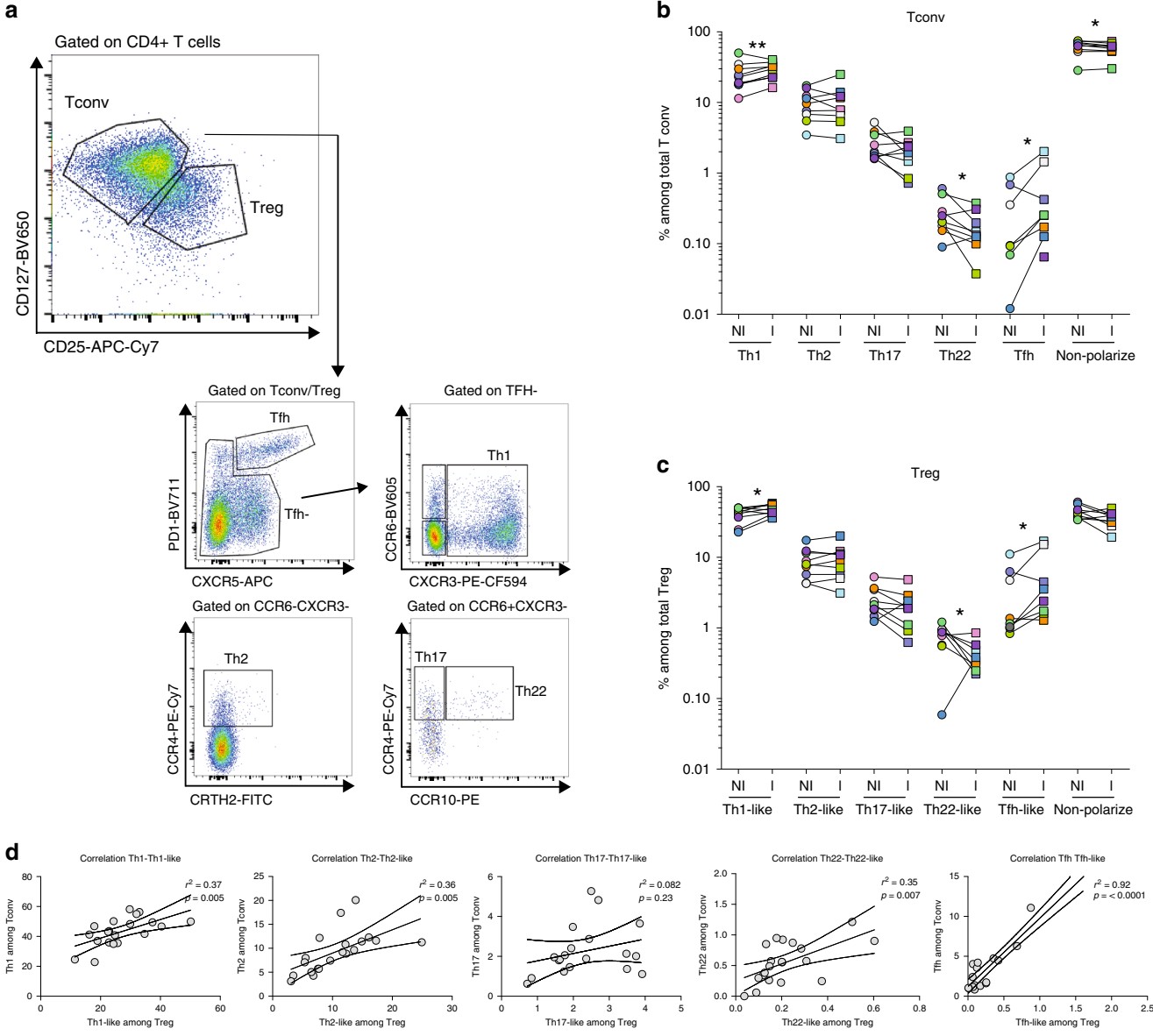

**Fig. 4 CD4+ T-cell chemokine receptor patterns in TDLNs.** Cell suspensions of TDLNs cells were stained for CD3, CD4, CD127, CD25, PD-1, CXCR5, CCR6, CXCR3, CCR4, CCR10, and CRTH2. **a** Representative flow cytometric analysis of CD4+ T-cell phenotype among gated CD4+ CD127+ CD25− (Tconv) or CD4+ CD127$^{low}$CD25$^{hi}$ (Treg) cells in TDLNs. **b**, **c** Frequency of cells expressing the indicated chemokine receptor combinations among **b** Tconv (Th1: $p = 0.0078$; Th22: $p = 0.0371$; Tfh: $p = 0.0391$; non-polarized: 0.0371in NI vs I TDLNs) and **c** Tregs (Th1-like: $p = 0.0195$; Th22-like: $p = 0.0391$; Tfh-like: $p = 0.0312$) in TDLNs. Wilcoxon matched-pairs signed rank test ($n = 10$) **d** Scatter plots and linear regression graphs of the correlation between the frequency of Th1 vs Th1-like, Th2 vs Th2-like, Th17 vs Th17-like, Th22 vs Th22-like, and Tfh and Tfr populations. *$P < 0.05$; **$P < 0.01$ in TDLNs ($n = 20$). The statistical significance of the correlation was determined using the Pearson's correlation. Non-polarized: CD4+ T cells negative for the assessed chemokine receptors.

frequency of TCR-β CDR3s in each sample (Fig. 5a, c and Supplementary Fig. 5a and 5c–d), indicates that Tregs were more clonally expanded than Tconvs in all tissues; and that the TCR repertoire of Tregs and Tconvs were less diverse in the tumor than in the TDLNs, suggesting an accumulation of tumor-specific CD4+ T cells in the tumor. Next, to analyze the TCR repertoire overlap among the three different tissues, for two patients we could identify the top 100 most expanded CDR3s of the Tregs or Tconvs present in the tumor (likely enriched in tumor-specific clones) and study their distribution in the NI and I TDLNs (Fig. 5b, d and Supplementary Fig. 5b). For Tconvs in each patient, 10.5% and 4.5% out of the 100 top tumor clones were found in the I TDLNs, and 1% and 3.4% of the 100 top tumor

clones were observed in the NI TDLN (Fig. 5b). For Tregs, 1.9% and 17.5% of the top 100 tumor clones were observed in the I TDLN and 2.6% and 1.6% of the 100 top tumor clones were observed in the NI TDLN (Fig. 5d). These results suggest that tumor-specific Tconvs and Tregs clones recirculate between the tumor and the TDLNs.

Tumor-specific Tregs can originate in the thymus (tTreg) or they can arise from conversion of Tconvs into "peripheral-induced" Treg, (pTreg)[36,37]. Peripheral Treg induction in TDLNs has been demonstrated in mouse models[38] but it has been poorly studied in humans[12,25]. To assess the pTregs presence, we evaluated the TCR repertoire overlap between Tregs and Tconvs. This analysis revealed that Tconvs and Tregs with shared TCRs

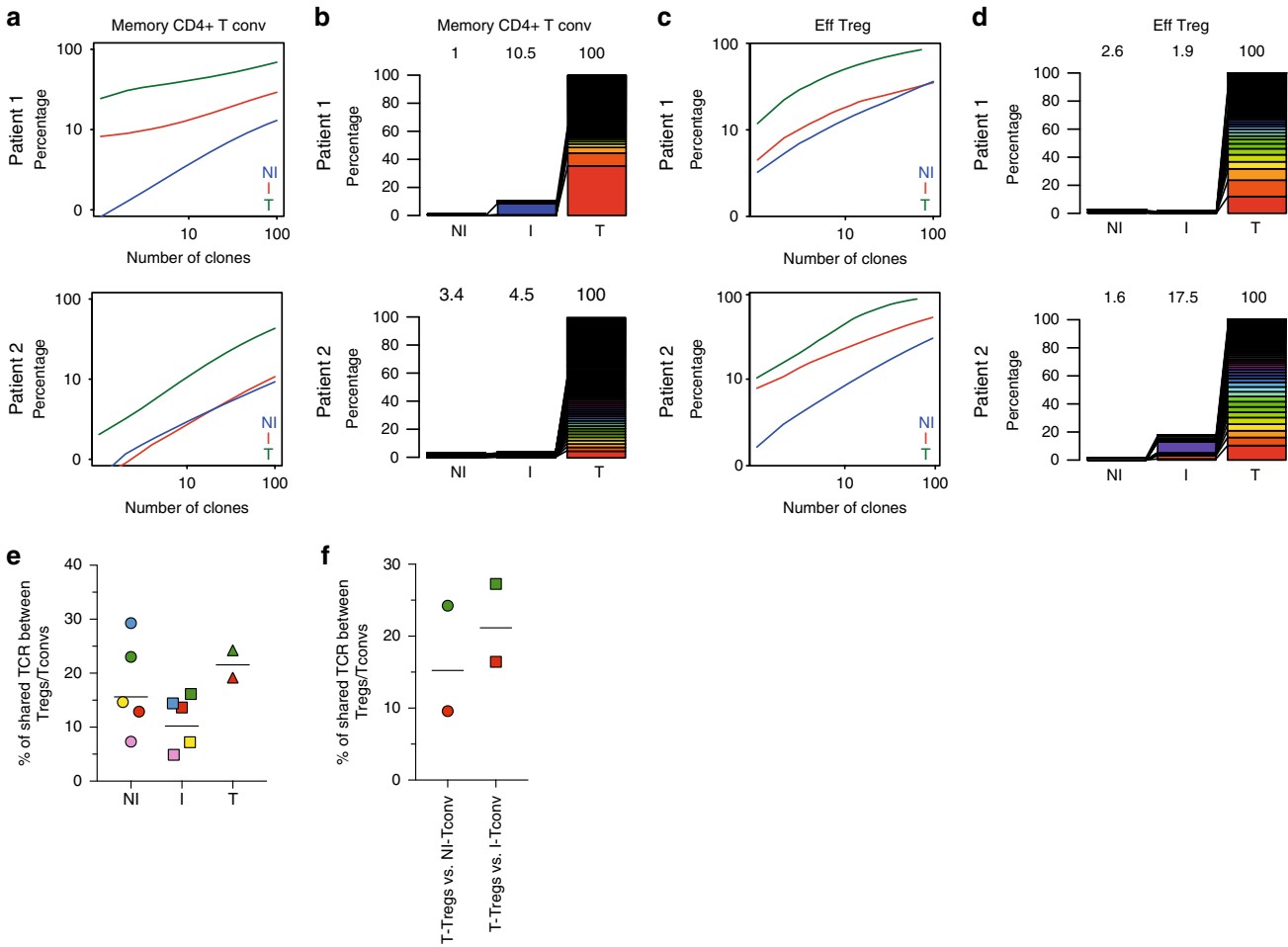

**Fig. 5 Analysis of the TCR repertoire of CD4+ Tconvs and Tregs from NI and I TDLNs, and tumor.** Next-generation sequencing-based high-throughput TCR-β CDR3 analysis of bulk sorted T-memory CD4+ Tconvs and Tregs from matched NI and I TDLNs and the primary tumor. Graphs **a**–**d** display the TCR-β CDR3 clones of memory CD4+ Tconv **a**, **b**; or effector Tregs **c**, **d**. Graphs **a**, **c** show cumulative frequencies of from NI TDLNs (blue), I TDLNs (red), and the corresponding primary tumor (green), for two representative patients. Stacked bar chart **b**, **d** depict the usage frequencies of the top 100 tumor TCR-β CDR3 sequences and their distribution in matched NI and I TDLNs for two representative patients. Numbers in the graphs indicate percentage of shown TCR-β CDR3 sequences out of total TCR-β CDR3 sequences in the sample. **e**, **f** TCR repertoire overlap between Tregs and Tconvs. **e** Shown is the number (%) of unique TCRs shared between Tregs and Tconvs among all Treg clones in TDLN (N = 5) and tumor (N = 2). **f** Shown is the number (%) of unique TCRs shared between all tumor Tregs and Tconvs present in the NI or I TDLNs (N = 2). Each symbol is a sample, and samples from the same patients are in the same color.

were present within every tissue, with a mean of 17.6% of the total Treg TCRs shared with the Tconvs in NI TDLNs, 14.3% in I TDLNs, and 21.7% in tumors (Fig. 5e and Supplementary Table 4). Moreover, tumor-Treg clonotypes were shared in an inter-tissue fashion with Tconvs, as 16.9% and 21.9% of the tumor-Treg clonotypes were also found within Tconvs present in NI and I TDLNs, respectively (Fig. 5f). These data uncover that a sizable proportion of Tregs in breast tumors are pTregs; however, do not allow concluding whether conversion occurs in one given organ or whether pTregs recirculate among the TDLNs and the tumor.

**Tregs from I TDLNs display unique transcriptomic features.** We performed RNA sequencing of bulk sorted Tregs and Tconvs obtained from matched NI, I TDLNs, and tumors (see methods). We observed 2985, 868, and 1800 differentially expressed genes (DEGs, fold change 1.2, p < 0.05) between Tregs and Tconvs in NI TDLNs, I TDLNs, and tumors, respectively (Fig. 6a, Supplementary Data 1). Pathway analysis of these three lists of DEGs identified a series of canonical biological pathways (Fig. 6b and

Supplementary Data 2). Among the top five canonical pathways, NI and I TDLNs shared Th1 and Th2 activation/polarization pathways and the "T-cell exhaustion signaling pathway", indicating chronic activation and polarization as main characteristics of the CD4+ T cells in TDLNs. Differently, tumor CD4+ T cells were principally associated with translational regulation ("Regulation of eIF4 and p70S6K Signaling" pathway), activation of the Inositol phosphate pathway, and signaling mediated by glucocorticoids; highlighting intrinsic differences with the lymph node residing cells.

Then, we studied the molecular features shared by Treg and Tconv ("common signature") analyzing the DEGs common between Tregs and Tconvs from all tissues; and to investigate tissue adaptation, we used the lists of DEGs uniquely expressed by Tregs or Tconvs of each tissue. The "common Treg signature" consisted of 121 genes (commonly upregulated in the Treg/Tconv comparison of the three tissues, "UUU" genes in the Venn diagram of Fig. 6a), included *FOXP3, IL2RA, TNFRSF9, TNFRSF1B*, and *ENTPD1* (Fig. 6c), and was significantly associated with "Regulatory T cells" and "IL-2 signaling

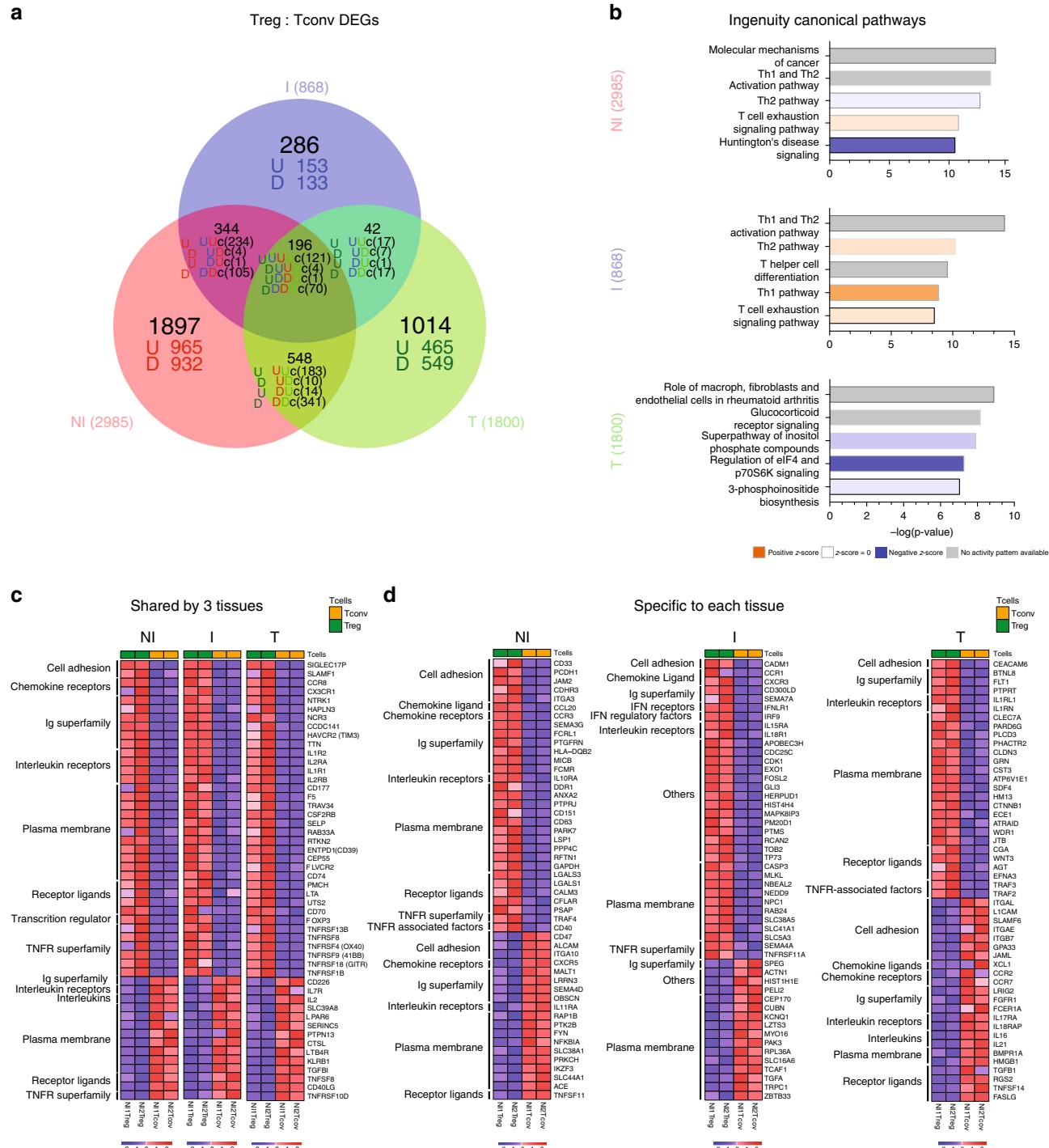

**Fig. 6 Transcriptional analysis of CD4+ Tconvs and Tregs from NI and I TDLNs, and tumor.** RNA-sequencing analysis of bulk sorted T-memory CD4+ Tconvs and Tregs from matched NI and I TDLNs and the primary tumor. **a** Venn diagram and **b** pathway analysis of differentially expressed genes (DEG, fold change 1.2, *p* < 0.05) between Tregs and Tconvs in NI TDLNs, I TDLNs and tumors. **c, d** Heatmaps showing 50 DEGs common to Treg and Tconvs from the three tissues **c** or specific to each tissue **d**. Red and blue indicate higher and lower expression, respectively. Family gene names are indicated (left). Cell adhesion: cell adhesion molecules. Ig superfamily: immunoglobulin superfamily domain containing. TNFR receptor superfamily: tumor necrosis factor receptor superfamily. TNFR-associated factors: tumor necrosis factor receptor-associated factors. IFN receptors: interferon receptors. IFN regulatory factors: interferon regulatory factors.

pathways" as well as "TNFR2 signaling pathway" (EnrichR, #1, Supplementary Data 3). The "common Tconv signature" was defined by 70 genes ("DDD" genes) including *IL7R, CD40LG,* and *IL2* (Fig. 6c), and was significantly associated with "CD4+ T cells", "FAS signaling pathway" as well as "TRAIL signaling

pathway" Fig. 6c (EnrichR, #2, Supplementary Data 3). Analysis of molecular signatures associated to each tissue revealed specific programs: (**i**) Tregs from NI TDLN were characterized by 965 genes ("U" genes), contributing to the "Antigen processing and presentation of exogenous peptide" and "DNA replication"

(EnrichR #3, Supplementary Data 3), reflecting basal replication of Tregs in the LNs; (ii) Tregs from I TDLN were characterized by barely 153 genes ("U" genes), associated to the biological pathways: "apoptosis", "IL-12-mediated signaling events", and Type III IFN signaling (EnrichR #4, Supplementary Data 3), likely reflecting Treg expansion and contraction in response to the local production of pro-inflammatory signals like IL-12 and IFNs; (iii) tumor Tregs uniquely displayed 465 genes ("U" genes) participating to a heterogeneous transcriptomic program comprising the pathways "Oxidative phosphorylation", "TNFR2 signaling", and "4-1BB signaling" (EnrichR #5, Supplementary Data 3), pointing out to a differential metabolic state in the tumor and signaling through TNFR2 and 4-1BB, previously associated to activated Tregs in the tumor[39,40]. Of note, tumor Tregs highly expressed *IL1RL1* and *IL1RN,* which have been associated to inhibition of IL-1-mediated inflammation[41,42]. These results suggest that Tregs from the tumor and the TDLNs express a common gene signature, but also exhibit distinct transcriptional cell fates underlying tissue-specific adaptation. Fig. 6c, d shows the expression levels of characteristic markers of Tconvs and Tregs cells selected by their influence on T-cell migration, function, or target potential. For example, *CCR8* is characteristic of Tregs across tissues (as previously described[12]), but other cell adhesion molecules and chemokine receptors are differentially associated to Tregs and Tconvs from the different tissues (Fig. 6d). In more detail, *SIGLEC17P* expression could be used by Tregs to circulate among all tissues; *CD33, PCDH1, JAM2, CDHR3,* and *ITGA3* would orchestrate migration/retention in NI TDLNs; *CADM1* in I TDLNS; and *CEACAM6* in tumor. Of note, *CEACAM6* expressed by tumor Tregs, has been associated with cancer progression[43]. Also, Tregs from NI ad I TDLN share *CD58, ITGAM, MCAM, CEACAM4, SELPLG,* and *HMMR* expression, which could ensure recirculation among TDNLs; and Tregs from I TDLN and tumor Tregs share *ICAM1* expression, which could be responsible for the migration among tissues with presence of tumor cells. Finally, Tconvs from the tumor differentially expressed higher levels of *IGAL* (LFA-1) than Tregs. These results underlie that different molecular cues attract and retain the Treg and Tconv cells to the TDLNs or to the tumor.

**CD80-expressing tumor Tregs correlate with bad prognosis**. To evaluate the clinical relevance of DEGs highly expressed in Tregs from TDLNs and tumors (Fig. 6a) we investigated the impact of these genes on tumor recurrence and survival using the breast cancer archive from the Cancer Genome Atlas (TCGA-BRCA, see methods). We first assessed *FOXP3* gene; the master transcription factor of Tregs, and observed that *FOXP3* mRNA expression level per se has no impact in overall survival (OS) and disease free-survival (DFS), as previously described[12] (Supplementary Fig. 6a–b). To identify genes which expression correlated with *FOXP3* across the TCGA-BRCA cohort, we calculated the gene: *FOXP3* mRNA ratio for all DEGs (see methods). We found 10 genes, in which mRNA levels were positively correlated with *FOXP3* reads ($r > 0.5$) and high mRNA:*FOXP3* ratios were associated with better OS and DFS, being *CD79A* and *TNFRSF13B,* two molecules associated with B-cell activation[41], the most significantly correlated ones (Fig. 7a–d and Supplementary Fig. 6c). Furthermore, we found a signature of three genes, namely *CD80, CCR8,* and *HAVCR2,* which positively correlated with *FOXP3* reads (Supplementary Fig. 6d) and which high mRNA:*FOXP3* ratios were significantly associated with poor OS ($p = 0,012$) and DFS ($p = 0.01$) (Fig. 7e, f). When considering these three genes independently, only the *CD80:FOXP3* mRNA high ratio was significantly associated with a worse OS and DFS (Fig. 7g, h). We confirmed the high expression of *CD80* on tumor Tregs at the

protein level by FACS (Fig. 7i and Supplementary Fig. 6g), but not in the I-TDLN because of the high variability among patients (Supplementary Fig. 6e). We found that tumor Tregs showed a higher frequency and MFI of CD80 than Tconv and peripheral Tregs from healthy donors (HD) (Fig. 7j). Of note, we detected CD80 expression in Tregs, both at the mRNA (RNA sequencing) and at the protein level (fluorescence-activated cell sorting (FACS)); as CD80 can be acquired by Tregs by trogocytosis[44] or transendocytosis[45,46], it is not possible to know whether the CD80 protein observed in Tregs has been synthetized by the Treg, has been acquired from the membrane of other cells, or both. These results indicate that CD80-expressing Tregs could define a subset of highly activated/suppressive Tregs associated with bad prognosis. These results emerge as potential therapeutic targets for breast tumor-associated Tregs.

## Discussion

Studies on Tregs in breast cancer showed that the proportion of blood FOXP3+ CD4+ T cells is increased compared with healthy donors[47,48] and that in the tumor high proportions of FOXP3+ CD4+ T cells—detected by immunohistochemistry (IHC)—are associated with a bad prognosis[14,15,25,49]. Data quantifying Tregs in breast cancer-TDLNs, is scarce[18], and has been mainly obtained by FOXP3 detection by IHC that do not allow the distinction between the suppressive CD4+ CD25+ FOXP3high T and the non-suppressive CD4+ CD25+ FOXP3low-T cells[12,18,19,50]. Our flow cytometry results establish that the high proportions of Tregs observed in TDLNs and in the primary tumor microenvironment constitute "bona fide" Tregs. Although Tregs can lose their phenotypic stability and functions in pro-inflammatory microenvironments[13,32,51], we found that even in the presence of activated IFN-γ-producing CD4+ T cells, Tregs in I TDLNs maintained their ex vivo suppressor capacity and had a stable phenotype (as judged by the low levels of cytokine production). Moreover, Tregs in TDLNs and in the tumor expressed high levels of CTLA-4, ICOS, GITR, and OX40, phenotype associated to a high suppressive capacity. It is noteworthy that although the Tconvs showed signs of dysfunction in the tumor, the suppressor Eff Tregs did not, highlighting the latter's ability to adapt to the tumor microenvironment. One explanation to this observation could be that Tregs are more proficient than Tconvs to survive and function in the presence of hypoxia, acidosis, and nutrient deprivation[52], which can be assumed to be more stringent in the tumor than in the I TDLNs, owing to differences in tumor burden.

Our observations have several clinical implications. The transcriptomic analysis of purely sorted Tregs and Tconvs identified CD80-expressing Tregs as a subpopulation of Tregs associated to bad prognosis in breast cancer, which represents a candidate for the design of immunoregulatory therapies, and merit further analysis to better understand its role in the biology of tumor-associated Tregs. Also, we found that Tregs from luminal breast cancer patients showed a distinct pattern of highly expressed costimulatory and co-inhibitory molecules, including PD-1, CTLA-4, ICOS, GITR, OX40, and CD39, underscoring the value of these molecules as targets for immunomodulation. As follows, the non-depleting, anti-CTLA-4 Ab Tremelimumab (but not the depleting Ab Ipilimumab) showed limited activity in metastatic melanoma and breast cancer;[24,53,54] what could be explained by its low capacity to deplete CTLA-4-expressing Tregs. Also, our data predict that depleting Abs that target ICOS, GITR, and OX40 (such as the depleting anti-OX40 Ab MEDI6469[55]) could be good candidate drugs for patients with luminal breast cancer.

We observed that in TDLNs, Tregs and Tconvs show a similar pattern of chemokine receptor expression, reinforcing the concept

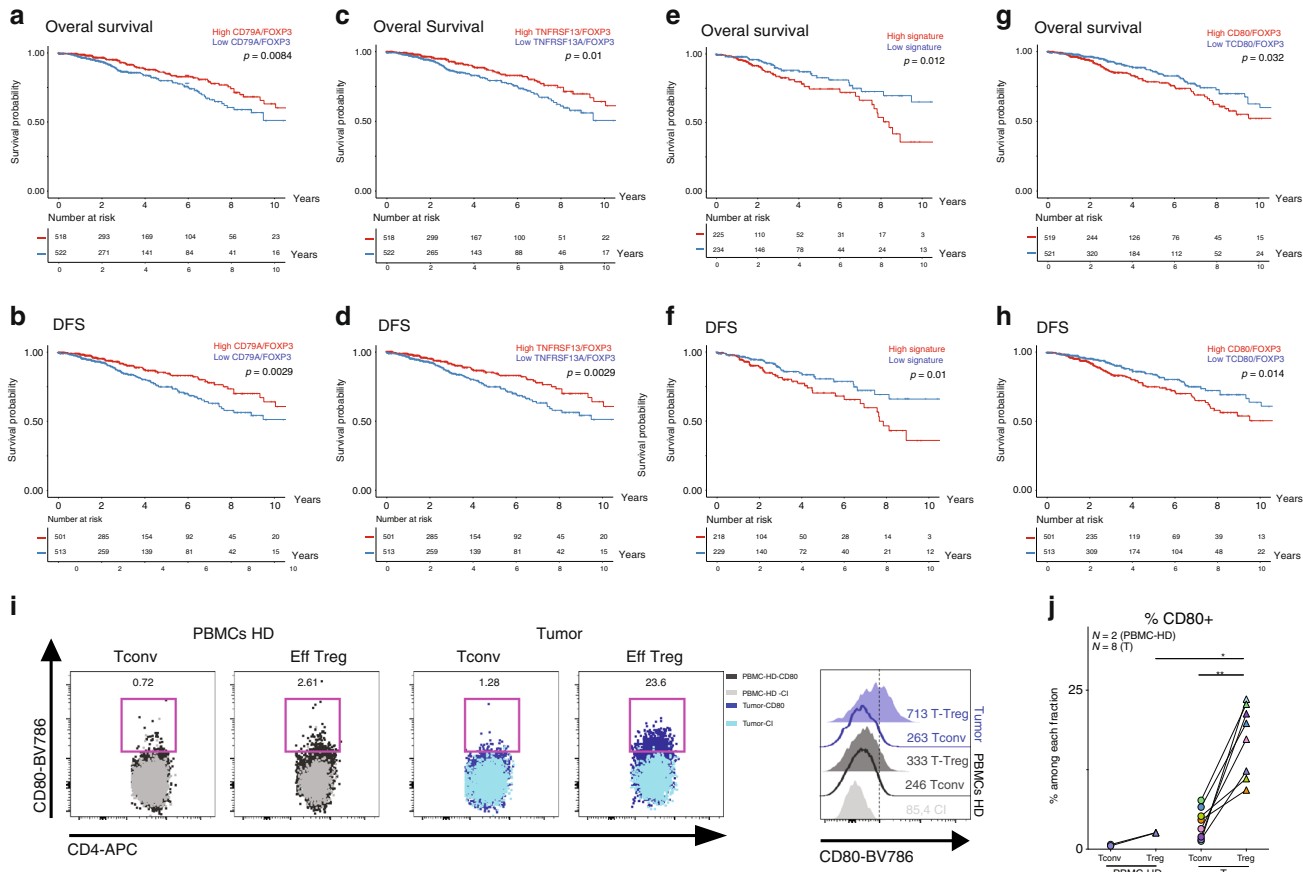

**Fig. 7 Clinical outcome of differentially expressed genes normalized to *FOXP3* expression. a–h** OS (upper panels) and DFS (lower panels) of patients with breast cancer stratified by "high" or "low" median intensity of the expression level of *CD79, TNFRSF13B, "CD80, CCR8,* and *HAVCR2* signature" and *CD80* normalized to *FOXP3* expression, in TCGA breast cancer data set. **i** Representative flow cytometric analysis of CD80 expression on Tconv and Treg cells from HD PMBCs (left panels) or tumor (right panels) and representative histogram of CD80 MFI levels in the different subpopulations. **j** Frequency of CD80+ CD4+ Tconvs or Tregs in HD PBMCs (N = 2) or tumors CD45+ cells (N = 8). PBMC-Treg vs tumor-Treg p = 0.044. Mann–Whitney test. Tumor-Tconv vs tumor-Treg p = 0.0078. Wilcoxon matched-pairs signed rank test, *p < 0.05, **p < 0.01.

that the ability of Tregs to maintain local immune homeostasis depends on their appropriate colocalization with Tconvs[13]. Along these lines, Redjimi et al.[56] also showed that the proportion of CXCR3+ Tregs in ovarian tumors was directly correlated to that of CXCR3+ Tconvs. In addition, in I TDLNs we detected a significant increase of Th1 cells compared with NI TDLNs, suggesting that nodal tumor invasion is associated to an IFN-γ-mediated T-cell response, which has been linked to potent antitumor responses[57,58]. Moreover, a rich CD4+ Th1 signature in breast tumors has been associated with a good prognosis[59]. Also, I TDLNs showed higher proportions of Tfh cells. Interestingly, Faghih et al.[8] identified Tfr and CD4+ T cells producing IFN-γ, IL-4, and IL-17 in breast cancer tumors, and concluded that upon metastasis in the TDLNs, the immune response becomes more inhibited.

In line with our results, other studies found higher levels of T-cell clonality in the tumors than in the blood and in juxtatumoral tissues (and in the TDLNs in our study), with shared clones found in tissues and tumors[12,25,35,60]. Assuming that the tumor-expanded clones are the tumor-specific ones, this observation underscores an ongoing antitumor response, leading to an accumulation of tumor-specific CD4+ T cells in the tumor. Furthermore, the detection of shared β-CDR3s between tumor-expanded T-cell clones and T cells in TDLNs indicates circulation between the tumor and the TDLNs, and underscores the potential contribution of TDLNs as source of tumor-specific T cells that

might be modulated by immunotherapies. As suggested by Zemmour et al.[61], Tregs with the same TCR have similar transcriptional traits, implying that Tregs with shared antigenic specificity from tumors and TDLNs may also share an early imprinted specific program that persists after priming and drive them to anatomical locations with common environmental cues. Furthermore, inter-tissue sharing of Treg clones could also be indicative of a loco-regional suppression mechanism. Finally, ~20% of the tumor-Treg β-CDR3s were shared with Tconv cells, pointing out that an important fraction of the Tregs could arise through peripheral induction or conversion of CD4+ Tconv cells. Along these lines, although Helios is not a marker exclusively for thymic Tregs, the high level of Helios expression in tumor and TDLN Tregs (Supplementary Fig. 2) could reflect in situ peripheral Treg differentiation. Although the use of the same TCR in different subsets is suggestive but not demonstrative of a common clonal origin, it has been recently shown in a mouse model[38] that monoclonal CD4+ T-cell may become effector, anergic, or Treg in the TDLNs as soon as 7 days after priming. Of note, in all the analyzed tissues Tregs were more clonally expanded that Tconvs, probably reflecting their more self-reactive repertoire.

We identified a group of 121 common genes that characterize Tregs in breast tumors, NI, and I TDLNs, and a list of distinctive genes associated to each location, pointing out their tissue adaptation. NI TDLN-Tregs are characterized by a

homeostatic proliferation signature, and I-TDLN-Tregs display a phenotype halfway between the NI and the tumor Tregs. Noteworthy, Tregs from NI TDLN are primed for antigen presentation, whereas Tregs from I TDLN tend to apoptose, what may enhance their suppressive capacity, as reported by Maj et al.[62]. Tumor Tregs are characterized by an activation signature and by metabolic changes, which may reflect their adaptation to the harsh metabolic conditions of the tumor microenvironment, characterized by hypoxia and acidosis, alterations in nutrient composition, like glucose restriction, high levels of reactive oxygen species (ROS), adenosine, and prostaglandins[63].

Among the specific tumor-Treg features, we observed a higher expression of *IL1RL1* (IL-33 receptor, *ST2*), which has been described to correlate with tumor number and size[51] and with inhibition of IL-1-mediated inflammation[41,42]. Of note, we observed an association of the *CD80/FOXP3* mRNA ratio with poor OS and DFS in breast cancer patients. It has been reported that T cells can acquire CD80 at an early state of activation[64]. In Tregs, the presence of CD80 at the protein level has been described linked to trogocytosis[44] or transendocytosis mediated by CTLA-4[45,46], and it has been associated as a mechanism of Treg-mediated suppression, as restriction of costimulatory ligand expression in dendritic cells inhibits CD28-mediated activation of T cells. CD80 has also been associated to Treg signatures of bulk transcriptomic data[65]. Only one recent work has also reported CD80 expression in activated Tregs[12] in autoimmune patients, and warrants further understanding on the role of CD80 in Treg biology.

Hypermutated cancers are viewed as potentially good responders to immunomodulators, owing to the activation of neoantigen-specific T cells in the tumor[57,58,66]. Breast cancers show low mutational loads, and patients with this cancer have been less included in clinical trials of immune checkpoint inhibitors than more-mutated tumors such as melanoma or lung tumors, yet some efficacy has been observed[23,24]. Adoptive T-cell therapy using neo-antigen specific autologous TILS combined with IL-2 and checkpoint blockade[67] can also be effective in these patients. Thus, combination of strategies boosting effector T cells with Treg-cell depletion should increase therapeutic efficacy of immunotherapies. Along these lines, our data provide information on the phenotype and function of Tregs in luminal cancer patients, and add "LN-T cells" as a dimension to be considered in the design of effective immunotherapies for cancer patients.

## Methods

**Clinical samples**. TDLNs and tumors were collected from 54 patients with luminal breast cancer having undergone standard-of-care surgical resection at the Institut Curie Hospital (Paris, France), in accordance with institutional ethical guidelines and informed consent was obtained. The protocol was approved by the Ethical Committee of Curie Institut ("Comité de la Recherche Institutionnel", CRI-0804-2015).

The study cohort included patients for whom samples of the NI and I TDLNs and the primary tumor were available. The patients' clinical and pathologic characteristics are summarized in Supplementary Table 1. Tumor metastasis in TDLNs was diagnosed using histologic and IHC techniques (cytokeratin positivity). Primary tumors were characterized by IHC screening of hormone receptors (estrogen receptor 1, encoded by *ESR1*, and the progesterone receptor, encoded by *PGR*), the tyrosine kinase cell surface receptor HER2 (also known as erbB-2 receptor tyrosine kinase 2, and CD340, encoded by *ERBB2*), and KI-67. Lymph node invasion by tumor cells was confirmed by Epcam/CD45 detection by flow cytometry.

**Samples and cell isolation**. Samples were obtained within a few hours after the primary surgery, cut into small fragments, and digested with 0.1 mg/ml Liberase TL (Roche) in the presence of 0.1 mg/ml DNase (Roche) for 20 min before the addition of 10 mM ethylenediaminetetraacetic acid. Cells were filtered on a 40-μm cell strainer (BD Biosciences).

**Phenotypic analysis of immune cell populations**. TDLNs and primary tumors were stained with the antibodies listed in Supplementary Table 5. Non-specific binding was blocked using anti-CD32 (Stem Cell). For intracellular staining, cells were fixed and permeabilized with fixation/permeabilization solution (eBiosciences), according to the manufacturer's instructions. All samples were acquired a few hours after surgery. Data acquired with a BD LSR-Fortessa flow cytometer were compensated, exported into FlowJo software (version 10.0.8, TreeStar Inc.), and normalized using Cyt MATLAB (version 2017b). To obtain an unbiased overview, we systematically reduced the flow cytometry data to two dimensions by applying t-distributed stochastic neighbor embedding (t-SNE, which displayed randomly selected live cells from TDLNs and primary tumors of all samples) in conjunction with FlowSOM clustering (which displayed all live cells from the three individual tissues of all samples)[68].

**Intracellular cytokine staining**. TDLNs were stimulated for 4 h with 100 ng/ml of PMA and 1 μg/ml ionomycin in the presence of GolgiPlug (BD Biosciences) for the final 3 h of culture. Cells were fixed and permeabilized with fixation/permeabilization solution (eBiosciences), according to the manufacturer's instructions. Cells were stained with FOXP3-Alexa488 (clone 236 A/E7, eBiosciences), IL-17-BV711 (clone BL168; BioLegend) and IFNγ-V450 (clone B27; BD Biosciences), and then analyzed on a Fortessa flow cytometer (BD Biosciences). The FACS data were analyzed with FlowJo software (version 10.0.8, TreeStar Inc.).

**CD80 staining**. Tumors from eight patients with breast cancer and 2 PBMC from heathy donors were stained twice first with surface and then with intracellular staining with CD80-BV786 (clone L307.4; BD Biosciences) and IgG1k-BV786 (clone X40; BD Biosciences) isotype.

**Treg suppression assay**. CD4+, CD25+, and CD4+ CD25− T lymphocytes were first isolated using the CD4+ CD25+ regulatory T-cell isolation kit (Miltenyi Biotec, 130-091-301) and then stained with CD25−PE (clone M-A251; BD Biosciences) and CD4-PE-CF594 (clone RPA-T4; BD Biosciences). CD4+ CD25− (Tconv) and CD4+ CD25high (Treg) cells were sorted by flow cytometry using a BD FACS ARIA II cell sorter. Tconvs were stained with CFSE (5 μM) (Life Technologies) and co-cultured with autologous Tregs at varying concentrations in a 96-well round bottom plate ($2.5 \times 10^4$ Tconvs per well) in the presence of Dynalbeads CD3/CD28 T-cell expander (Life Technologies) at a ratio of 10 cells/bead. Cells were incubated at 37 °C for 4 days in Roswell Park Memorial Institute 1640 Medium (Life Technologies) supplemented with 10% AB-human-serum and 1% penicillin–streptomycin, and were then analyzed using flow cytometry (BD LSR-Fortessa). Proliferation index was calculated with FlowJo Proliferation Tool (TreeStar Inc.).

**Cell sorting for RNA and TCR sequencing**. Matched NI, I, and TDLNs and primary tumors from patients with breast cancer were processed for RNA and/or TCR sequencing. T cells were first isolated using the Pan T Cell isolation kit (Miltenyi Biotec, 130-096-535). Non-specific binding was blocked using anti-CD32 (Stem Cell), and cells were stained with CD25−PE (clone M-A251; BD Biosciences), CD4-PE-CF594 (clone RPA-T4; BD Biosciences), CD8-Alexa700 (clone 3B5; Life Technologies), CD27-APC (clone L128; BD Biosciences) or CD45RA-PE-Cy7 (clone HI100; BD Biosciences) and then with 4′,6-diamidino-2-phenylindole (DAPI) LIVE/DEAD stain. CD4+ CD45RA-CD25− (Memory CD4+ Tconv), CD4+ CD45RA-CD25high (Memory Treg), were sorted by flow cytometry in a BD FACS ARIA II cell sorter, with a purity of 98–99. Cells were collected and lysed with TCL buffer (Qiagen) with 1% of β-mercaptoethanol and stored at −80 °C until subsequent analysis. RNA was isolated using a Single Cell RNA purification kit (Norgen), including RNase-Free DNase Set (Qiagen) treatment. The RNA integrity number was evaluated with an Agilent RNA 6000 pico kit. All samples were assessed according to the manufacturer's instructions.

**TCR sequencing**. The RNA was extracted from all the sorted cells. After purification, all the RNA was used for the reverse transcription step. A specific reverse transcription reaction was performed with constant TCRβ region primers (see below for the primer sequence) coupled at the 5′-end to the common sequence 2 (CS2) TACGGTAGCAGAGACTTGGTCT using SuperScript IV (ThermoFisher). After synthesis, cDNA was cleaned using Agencourt RNAclean XP kit (Beckman Coulter) according to the manufacturer's instructions, and then eluted in 40 μl of RNAse-free water (Ambion). We used three PCR steps for cDNA amplification and barcoding. For PCR reaction 1, we used the same reverse constant primers as for reverse transcription, and we used previously described forward TCR sequence primers[69]. For the PCR reaction 1, the concentration of each TCR region primer was 0.2 μM. We performed 17 cycles for the first PCR step to keep it in the exponential phase allowing > 10,000-fold amplification and an unbiased representation of all the TCR β transcripts, using the following cycling conditions: 95 °C for 3 min, 90 °C for 30 s, 63 °C for 30 s, and 72 °C for 30 s. After the first multiplex PCR, cDNA was cleaned using an Agencourt AMPure XP kit (Beckman Coulter) according to the manufacturer's instructions, and eluted in 40 μl of RNAse-free water. After beads-based purification, 1/160th of the purified product was used as template for the second PCR allowing a good representation of the initial template.

For PCR reaction 2, we performed two distinct, seminested multiplex PCRs for TCRβ, using previously described TCR variable region primers[69] (Supplementary Table 6) coupled at the 5′ end to the common sequence 1 (CS1, ACACTGAC-GACATGGTTCTACA) primer. We used the first PCR product as template, and seminested PCRs were performed in a 20 μl final reaction volume using GoTaq G2 Hot Start polymerase (Promega) according to the manufacturer's instruction. We performed 30 cycles with the following cycling conditions: 95 °C for 3 min, 90 °C for 30 s, 63 °C for 30 s, and 72 °C for 30 s. The cDNA was cleaned using an Agencourt AMPure XP kit according to the manufacturer's instructions, and resuspended in 40 μl of RNAse-free water. Barcoding and pair-end addition steps for Illumina sequencing were performed in a third PCR reaction using PE1_CS1 forward primer and PE2_barcode_CS2 reverse primer (Fluidigm) at 400 nM. For this final PCR, we used the second PCR product as the template in a 20 μl final reaction volume using Platinium Taq DNA Polymerase High Fidelity (Thermo-Fisher), with the following conditions: 94 °C for 10 min, 94 °C for 30 s, 60 °C for 30 s, 68 °C for 4 min, and 68 °C for 3 min. The cDNA was cleaned using an Agencourt AMPure XP kit, according to the manufacturer's instructions. Each PCR product had a unique barcode and a Fluidigm paired-end sequence that enable sequencing on an Illumina Miseq system. The CS2 TCR primer sequence is TACGGTAG-CAGAGACTTGGTCTTACCAGTGTGGCCTTTTGGGTGTG, with the common sequence is indicated in bold. We adapted the number of reads per sample according to the initial number of cells.

**TCR-sequencing analysis**. MiXCR[70] (version 2.1.5) was used with its default parameters to extract and quantify CDR3 sequences from raw TCR sequence data. Using the MiXCR output, TCRβ clones were defined according to the identified V, J, and CDR3 sequences. To reduce noise, we first filtered out clones with fewer than three counts and then selected the 90% most strongly expressed TCRβ clones for each sample. For the table 4, the percentage of shared clones was calculated as: number of shared clones/total number of Treg clones × 100, for each tissue.

**RNA sequencing**. Retrotranscription was carried out with SMARTv4 low input kit (Takara). Barcoded Illumina compatible libraries were generated from 5 to 10 ng of DNA of each sample, using Nextera XTP reparation Kit. cDNA was generated using SMART-seq version 4 low input kit (Takara), Barcoded Illumina compatible libraries were generated from 5 to 10 ng of DNA of each sample, using Nextera XTP reparation Kit. Libraries were sequenced on an Illumina HiSeq 2500 using 100 bp paired-end mode, 25 millions reads per sample.

**RNA-sequencing analysis**. FASTQ files were mapped to the ENSEMBL Human (GRCh38/hg38) reference using Hisat2 and counted by featureCounts from the Subread R package. Read count normalization and groups comparisons were performed by three independent and complementary statistical methods: Deseq2, edgeR, LimmaVoom. Flags were computed from counts normalized to the mean coverage. All normalized counts < 20 were considered as background (flag 0) and ≥ 20 as signal (flag = 1). P50 lists used for the statistical analysis regroup the genes showing flag = 1 for at least half of the compared samples. The results of the three methods were filtered at $p$ value ≤ 0.05 and folds 1.2 compared and grouped by Venn diagram. Cluster analysis was performed by hierarchical clustering using the Spearman correlation similarity measure and average linkage algorithm. Heatmaps were made with the R Version 1.1.463 (Cluster and Tree Conversion) and imaged by Java Treeview software. Functional analyses were carried out using Ingenuity Pathway Analysis (IPA, Qiagen), Hugo Gene Nomenclature Committee (HGNC) and EnrichR.

**Survival analysis**. To assess the association of gene expression level with the overall survival (OS, defined as time to death) and DFS (defined as time to recurrence or death), we used legacy bulk RNA-seq data provided by The Cancer Genome Atlas (TCGA-BRCA, TCGAbiolinks package) obtained from 562 Luminal A, 209 Luminal B, 190 Basal, and 82 Her2+. As OS and DFS of *FOXP3* mRNA expression level per se is not a good read out of the well known contribution of Tregs to tumor scape, we used a bioinformatics strategy, as proposed by Plitas et al.[12]. In brief, first, for the survival analysis we selected from the list of DEGs obtained from Fig. 6a, only those genes that positively correlated ($R > 0.5$ and counts> 20 RPM) with FOXP3 reads per million of mapped reads (RPM). Then, to evaluate the gene ratios, DEG RPMs were normalized by *FOXP3* RPM and the obtained DEGs:FOXP3 proportions were used to segregate the breast cancer patients in two groups (low and high) based on the median expression level. Statistical significance of the curves (OS and DFS) stratified by the above-mentioned groups was determined using a Log-rank test (R package survival).

**Statistical analysis**. Statistical significance was determined using Wilcoxon's and Mann–Whitney's test for paired and non-paired samples respectively and the Pearson correlation coefficient. All statistical analyses were performed with Prism 6 software. Correlations between the immunological parameters and clinical/pathological data were probed in a Pearson/Spearman correlation analysis (R software environment, version 3.4.0).

**Reporting summary**. Further information on research design is available in the Nature Research Reporting Summary linked to this article.

## Data availability

RNA-seq data that support the findings of this study have been deposited in ArrayExpress with the E-MTAB-9112 accession code RNA-seq. TCR-seq data that support the findings of this study have been deposited in NCBI Gene Expression Omnibus with the GSE115545. The authors declare that all other data supporting the findings of this study are available within the paper and its supplementary information files.

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

## Acknowledgements

We thank Vassili Soumelis, Emanuela Romano, and Clotilde Théry for discussions and critical review of the manuscript, Zofia Maciorowski and the Institut Curie Flow Cytometry facility for cell sorting, André Nicolas, Ana Tereza Nadan, and Nathalie Amzallag for technical assistance, Phil Cheng for TCGA data analysis support and the Institut Curie NSG platform facility. We also thank the TCGA for providing breast cancer RNA-seq and clinical data sets. This work was funded by the Institut National de la Santé et de la Recherche Médicale (INSERM); the Association pour la Recherche sur le Cancer (ARC); the Institut Curie; the Agence Nationale pour la Recherche (ANR Emergence program); the Institut National du Cancer (INCa), IGR-Curie 1428 Clinical Investigation Center, Labex DCBIOL (ANR-10-IDEX-0001-02 PSL* and ANR-11-LABX0043), SIRIC (INCa-DGOS-Inserm_12554, projets 2011–2017:INCa-DGOS-Inserm_4654). N.G.N. received a fellowship from Ligue Nationale Contre le Cancer and the University Research Priority Program (URPP).

## Author contributions

N.G.N., J.T.B., C.S. O.L., and E.P. designed experiments and wrote the manuscript; N.G.N., J.T.B., R.N.R., L.L.N., J.B., W.R., P.D., S.V., and L.P. performed experiments; D.M., A.T.N., A.V.S., X.S.-G., and D.L. provided patient samples and clinical information; M.M. and D.L. provided logistic support and managed patient databases, W.R., N.G.N., N.C. and C.D.A. provided bioinformatics assistance, C.S., B.B., R.N.R., and S.A. provided critical design input and edited the manuscript; and E.P. conceived the study.

## Competing interests

The authors declare no competing interests.

**Additional information**

