## [Peer Review File · Nature Communications]

Reviewers' comments:

Reviewer #1 (Remarks to the Author):

Much work has been carried out using high throughput methods, such as CyTOF and single-cell RNA sequencing, to uncover the heterogeneity of tumor-infiltrating Tregs. In this manuscript, Gonzalo and colleagues have taken a step further by investigating Tregs within tumor-draining lymph nodes which either have (I) or have not (NI) been invaded by metastatic tumor cells. They claim that CD80-positive Tregs are prevalent in patients with poor clinical outcome, hence could serve as a prognostic marker. Although this study is, perhaps, intended to be purely descriptive, the authors should conduct further analyses as recommended below to better understand the relevance of certain characteristics (e.g. shown by DEGs) related to tumor Tregs and those in the dLNs.

1. In Figure 1E, the colour representing Treg cluster should be made more distinct given that it is the only population significantly different between NI and I dLNs.

2. The data in Figure 2 largely reflects other reports in the current literature that show increased expression of activation and inhibitory molecules in tumor Tregs. The increase in PD-1 and ICOS in effector Tregs of I dLNs compared to NI dLNs is, therefore, not surprising. Given that the expression of both molecules (Fig. 2B and D) appear to be bimodal in NI and I dLNs, the authors can further analyze eTregs with single or double expression of PD-1 and ICOS. They may very well find expansion of a particular population (e.g. PD-1+ICOS+ versus PD-1+/-ICOS+/-) in the I dLNs.

3. Chemokine receptors are certainly useful in identifying Th1-like, Th2-like ...etc Tcons and Tregs. Did the authors find any increase in the canonical molecules associated with function, for example T-bet and Eomes for Th1-like, Gata3 for Th2-like and Bcl6 for Tfr in transcriptome analysis?

4. In Fig 3C where the suppressive functions of Tregs from I dLN and tumor were assessed, a more meaningful comparison would be Tregs from I and NI dLN. According to RNAseq analysis, Tregs from NI dLN, I dLN and tumor have discrete biological pathways. For example, NI dLN Tregs are primed for antigen presentation whereas I dLN Tregs tend to apoptose. The latter may enhance Treg suppressive capacity as reported in *Nat Immunol.* 18, 1332–1341 (2017). Some application of knowledge gained from RNAseq analysis to Treg functional assay (e.g. addition of Th1/Th2 cytokines, 4-1BB agonist) would be insightful. For instance, 4-1BB activation could favor tumor Tregs more so than dLN Tregs to suppress.

5. Overlapping TCR repertoire between tumor and dLN T-cells supports shuttling of T-cells between the sites. Did the authors also assess cell trafficking molecules like LFA-1 especially in I dLNs?

6. Common TCR repertoire may be evidence of pTreg development in the tumor and dLN. Nevertheless, Helios expression in Tregs at the various locations should be mentioned (as in Supp Fig.2a). Although Helios is not a marker exclusively for thymic Tregs, its differential expression indicates the possible level of in situ pTreg differentiation in tumor and dLN.

7. CD80 was found to be upregulated in Tregs associated with reduced OS and DFS. This was confirmed at the protein level for Tcons and Tregs (Fig. 7I and J). However, since the focus of this paper is on Treg accumulation within draining lymph nodes invaded by tumor cells, the expression (either RNA or protein) of CD80 in Tregs from NI dLN, I dLN and tumor ought to be presented as well. Could the difference in CD80 expression between HD and cancer patients be detected in PBMC Tregs? Lastly, as the authors deem CD80+ Tregs to be highly stable and activated, it is worth assessing the effect of CD80 blockade on Treg stability and survival. This could be done in vitro with tumor Tregs alone or co-cultured with Tcons.

Minor points:

1. The word 'revealed' is misspelled several times as 'reveled'.
2. A brief description of tumor environment (hypoxia?) in breast cancer tissue which Tregs adapt to can be added in discussion.
3. CD79a and TNFRSF13B are conventional B-cell receptors. Can the authors speculate reasons for their possible role/s in rendering Tregs 'fragile', particularly for TNFRSF13B which has been shown to be present on T-cells, contributing to their activation?

Reviewer #2 (Remarks to the Author):

This study by Gonzalo and colleagues, which is well designed and nicely executed, explores and further characterizes the Treg compartment in ER+ breast tumors and their draining lymph nodes. The authors make some interesting observations that will need further functional validation to confirm their potential clinical impact. On the whole, their observations are rather incremental as (contrary to their claim) flow cytometry based studies have been published before, showing accumulation of CD25hi FoxP3hi(Ki67+) effector Tregs in tumor-involved nodes (and -to a lesser extent- even in non-involved nodes, as compared to healthy breast-draining nodes, see van Pul et al J Immunother Cancer 2019) as well as in tumors, wherein Plitas and colleagues showed functional suppression, PD-1 expression, and also performed transcriptional analyses, comparing Tregs in breast tumors to Tregs in normal breast tissue and PBMC (Immunity 2016). While the authors make some interesting observations in terms of potentially novel therapeutic targets (CD80 on Tregs), these are mostly observed in tumors rather than lymph nodes and remain to be functionally confirmed as viable targets to reduce Treg numbers or activity.

Specific comments:

- 1) Why was the actual suppressive activity of Tregs in NI lymph nodes not tested?
- 2) While Tregs from I TDLN and T show similar in vitro suppressive activity, the proliferative Tconv capacity is reduced in T: how do the authors explain this? They claim their findings from performed suppression assays point to a "stable" Treg phenotype in I TDLN, but since this is an in vitro assay with isolated Tregs, it is questionable that this is representative for their in vivo activity. Also, an absence of IFN-gamma production is no guarantee of active suppression. Please discuss.
- 3) It seems questionable that shared TCR sequences between Tconv and Tregs should be taken as incontrovertible evidence of Treg conversion from Tconv. It could also just mean shared specificity between separate Treg and Tconv clones. Please discuss.
- 4) Based on CCR/CXCR expression patterns, the authors claim a Th1-like profile for Tregs in I TDLN, whereas they have demonstrated an absence of IFN-gamma expression. So, what exactly do they mean by "Th1-like" -doesn't it rather demonstrate a shared migrational imprinting program with effector (Th1) Tconv, like previously described by the team of Drs. Sallusto and Lanzavecchia? NB: the used gating strategy of CD127-CD25+ here will not just select for Tregs but will also still comprise FoxP3+ activated Th cells.
- 5) Their use of CD80 and CD79a transcript levels "normalized" to FoxP3 (by using a ratio) seems odd...how does this guarantee actual co-expression? This would require scRNAseq?
- 6) Just an association with improved OS for e.g. Treg/CD79a, does not necessarily identify them as fragile Tregs: please rephrase and avoid such overstatements without functional/phenotypic confirmation.
- 7) Was there a relationship between Tregs and the activation state of Tconv, including CD8+ effector T cells in NI TDLN, I TDLN and T?
- 8) The observation of the presence of CD80+ Tregs in T is interesting: what about in NI TDLN and I TDLN? A possible relationship to OS warrants discussion of a possible underlying hypothesis. The authors mention CTLA4-mediated trans-endocytosis but don't refer to the paper in question: it

certainly deserves more discussion in relation to their identification of CD80+ Tregs; in this regard, how certain can they be of membrane vs. i.c. expression of CD80, as they apparently performed staining both before and after permeabilisation?

9) Finally, was there a relationship between Treg frequencies/activation state and 1) distance of TDLN to tumor, 2) size of the mets in I TDLN, 3) HR and Her-2/neu status: it is a pity the study only included patients with HR+ tumors. Certainly, in terms of impact for possible immunotherapy applications, the inclusion of triple negative tumors and their TDLN would have been highly relevant.

Reviewer #3 (Remarks to the Author):

It's true that detailed molecular and functional characterization of T cell subsets in draining lymph nodes of breast cancer and other cancer types is limited. Gonzalo et al. provides a relatively comprehensive profile of CD4 T cell subsets, mostly focusing on CD4 conventional and regulatory T cells (Treg), in matched tissues of positive and negative draining lymph nodes and lumina breast tumors. This study combines information of cellular composition, protein marker expression, bulk RNA transcription and TCR repertoire analysis. The strength of the study is comparative analysis on matched tissues of negative lymph nodes, positive lymph nodes and tumors in the same individual, which is definitely welcome and would provide useful information to the community. The authors find Treg is the only cell subset with higher proportion in invaded lymph nodes than the non-invaded ones, among all cell subsets they investigate. These Tregs in invaded lymph nodes express higher level of immune checkpoint molecules, reserve activating suppressive function and follow a similar polarization pattern as conventional T cells (Tcon). TCR repertoire result supports a proportion of conversion from Tcon to Treg. Bulk RNA sequencing identifies common Treg transcription signatures and tissue specific signatures, in which CD80 expression correlates with poor survival from TCGA samples. The whole study is well designed, and multiple state-of-art technologies are used. Most conclusions can be supported by the data, though some claims lack solid evidences. Overall, the story starts from an attractive finding (accumulative Treg cells in invaded lymph node) and provides rich information, however, the whole story seems to lack some striking highlights. Some important points mentioned are either consistent findings or lack further evidences. I will elaborate some points as follows.

1. L109-L111: "almost each" is an over statement. Treg proportions are lower in 6 out of 14 samples, and N=14 should be added in Fig. 1F.
2. L435 I cannot find supplementary table 8
3. L219 "that" should be "than"
4. There are some problems with the TCR repertoire analysis. Firstly, the authors miss plenty of important details in the method section, which would leave obstacles for the reproducibility of the experiment. No information is provided about the quantity of input RNA and the correspondent cell number, for the RT reaction. Are all the sorted cells used? Three PCR steps are used for the TRB library construction, but no input DNA quantity is mentioned for each step. The key is whether all the sorted cells listed in supplementary table 3 are processed to sequencing. If yes, it seems the produced sequencing reads are not enough to cover all the cells (especially for the conventional CD4), which will underestimate the number of clones. The authors haven't used any normalization method, which would make the number of clones being not representative of the number of sorted cells. There are three major conclusions in this section. 1) clonal expansion: tumor>invaded lymph node>non-invaded lymph node; 2)trafficking between tumor and lymph node; 3) conversion between Tregs and conventional T cells; For the first two points, the authors use the top 100 clones to perform the comparison, which won't be influenced too much by the problem I mentioned above. However, for point 3, the authors are not stringent to use all the clones in the data to calculate the shared percentage between Tregs and conventional T cells. Further, when calculating the percentage of shared clones, the authors should indicate whether the frequencies of clones are taken into consideration.

5. L235, Could the authors provide some biological significance for the inter-tissue sharing?
6. L381-384, when the authors discuss the sharing of TCR clones between tumor and TDLN they are preferable to mention previous reports (such as reference 83), though it's interesting to see similar result in sorted conventional and regulatory T cells.
7. CD80:FOXP3 ratio are identified to be associated with worse survival from the TCGA breast cancer data. As one of the DEGs between regulatory and conventional T cells, is CD80 a "common signature" or adapted to certain tissues?
8. CD80 is conventionally thought to be expressed in dendritic cells and other antigen presenting cells. When the authors discuss CD80 expressing regulatory T cells in tumor and use FACS to validate it, they shouldn't neglect previous reports about acquisition of CD80 to T cells upon activation (DOI: <https://doi.org/10.4049/jimmunol.169.11.6162>, <https://doi.org/10.1038/cmi.2011.62>). At least they are preferred to discuss why they think CD80 are expressed by T cells rather than being acquired from antigen presenting cells.
9. Inconsistent use of "tables" and supplementary tables"

Point-by-point response

Reviewer #1

Much work has been carried out using high throughput methods, such as CyTOF and single-cell RNA sequencing, to uncover the heterogeneity of tumor-infiltrating Tregs. In this manuscript, Gonzalo and colleagues have taken a step further by investigating Tregs within tumor-draining lymph nodes which either have (I) or have not (NI) been invaded by metastatic tumor cells. They claim that CD80-positive Tregs are prevalent in patients with poor clinical outcome, hence could serve as a prognostic marker. Although this study is, perhaps, intended to be purely descriptive, the authors should conduct further analyses as recommended below to better understand the relevance of certain characteristics (e.g. shown by DEGs) related to tumor Tregs and those in the dLNs.

1. In Figure 1E, the colour representing Treg cluster should be made more distinct given that it is the only population significantly different between NI and I dLNs.

Answer: As suggested by the reviewer, for a better visualization of the differences in figure 1E, we have changed the color of the Treg cluster to a more visible one (red). New updated figure 1_1.

2. The data in Figure 2 largely reflects other reports in the current literature that show increased expression of activation and inhibitory molecules in tumor Tregs. The increase in PD-1 and ICOS in effector Tregs of I dLNs compared to NI dLNs is, therefore, not surprising. Given that the expression of both molecules (Fig. 2B and D) appear to be bimodal in NI and I dLNs, the authors can further analyze eTregs with single or double expression of PD-1 and ICOS. They may very well find expansion of a particular population (e.g. PD-1+ICOS+ versus PD-1+/-ICOS+/-) in the I dLNs.

Answer: As suggested, we have analyzed the single and double expression of PD1 and ICOS on Eff Tregs from NI and I TDLNs and also the tumors. We observed that in I LN there is an increased frequency of PD-1+ICOS+ Tregs ($p < 0.05$), compared to NI TDLNs. This increase is accompanied by a decrease in PD-1-ICOS- Tregs ($p < 0.01$). We have now added these results in Figure 2E and described them in the results section page 4 L136.

This panel has now been added as panel E in Figure 2

3. Chemokine receptors are certainly useful in identifying Th1-like, Th2-like ...etc Tcons and Tregs. Did the authors find any increase in the canonical molecules associated with function, for example T-bet and Eomes for Th1-like, Gata3 for Th2-like and Bcl6 for Tfr in transcriptome analysis?

Answer: We studied by FACS the expression of T-bet and GATA3 in the different CD4+ T cell populations. Examples of FACS stainings and quantification have now been included in the supplementary figure 4 C and D, and added results are described in the result section page 5, L209.

This panel has now been added as Supplementary figure 4 C and D

FACS staining of Rorgt and Bcl-6 was not interpretable due to very few positive cells similar to FMOs (not shown).

In the transcriptomic data, the gene expression level of T-bet, Eomes, Gata3 and Bcl6 were either low or showed dispersion among the samples, rendering a difficult interpretation. We have included them here below, for reviewer perusal.

Gene	niLN_Treg_512	iLN_Treg_512	niLN_Treg_401	iLN_Treg_401
BCL6	33,3	14,6	19,4	28,5
GATA3	198,2	441,4	176,1	155,3
EOMES	9,7	5,7	1,6	318,8
TBX21	34,0	34,2	40,7	117,2
Gene	niLN_Treg_512	iLN_Treg_512	niLN_Treg_401	iLN_Treg_401
FOXP3	438,8	563,5	846,7	328,6

4. In Fig 3C where the suppressive functions of Tregs from I dLN and tumor were assessed, a more meaningful comparison would be Tregs from I and NI dLN. According to RNAseq analysis, Tregs from NI dLN, I dLN and tumor have discrete biological pathways. For example, NI dLN Tregs are primed for antigen presentation whereas I dLN Tregs tend to apoptose. The latter may enhance Treg suppressive capacity as reported in Nat Immunol. 18, 1332–1341 (2017). Some application of knowledge gained from RNAseq analysis to Treg functional assay (e.g. addition of Th1/Th2 cytokines, 4-1BB agonist) would be insightful. For instance, 4-1BB activation could favor tumor Tregs more so than dLN Tregs to suppress.

Answer: We agree with the reviewer on the relevance of this comparison, and we have previously envisioned doing such experiment. However, the size of the NI TDLN that the pathologists provide is really small, as *the NI LNs are quasi exclusively devoted to the clinical evaluation for diagnosis and staging of the patients*. The samples we have received along the years that took us to do this work were roughly sufficient to do phenotype and function analysis of unsorted T cell populations, and we could only do the RNAseq from sorted Treg cells from very few patients with occasional bigger NI LNs. Thus, we are unable to perform this experiment in a reasonable timeframe.

We thank the reviewer for highlighting the observation regarding the apoptotic state of Tregs in the I-LN. Based on the suggested manuscript “Nat Immunol. 18, 1332–1341 (2017)”, we have added two novel pieces of data, as follows:

1) we interrogated our RNAseq data about the Oxidative Stress Induced Gene Expression Via Nrf2 (BIOCARTA_ARE NRF2_PATHWAY). We found: ii) that differentially expressed genes between Tregs and Teff from I-LN were enriched in pathways related with DNA damage via ATM/ATR system, which has been associated with NRF2 inhibition (10.7750/BioDiscovery.2015.15.1, 10.1073/pnas.1207718109).

We have now included this new result in supplementary table 7 #EnrichR4 and commented the article in the discussion section, Page 9, starting in L425.

2) we quantified CD39 expression on Tregs from the different tissues. As shown in the Figure below tumor and I LN's eff Tregs expressed higher mean fluorescent intensity of CD39 than NI LN counterparts.

We have now incorporated this novel information in the revised version of the manuscript as follows:

- the figure above was incorporated as Figure S2B

- in the results section, page 4 L153, we added the following sentence:

“The higher CD39 expression in tumor and I LN Tregs than the NI LN counterparts (Figure S2B), indicates that the presence of tumor cells triggers the activation of Tregs and imprints a potential higher suppressive function.”

5. Overlapping TCR repertoire between tumor and dLN T-cells supports shuttling of T-cells between the sites. Did the authors also assess cell trafficking molecules like LFA-1 especially in I dLNs?

Answer: Our results from the transcriptomic analysis shown in Figure 6D, indicated that some adhesion molecules were upregulated in Tregs from the three sites, like SIGLEC17P, which could be a molecule used by Tregs to circulate among all tissues. Other adhesion molecules were only upregulated in Tregs from the NI TDLNs (like CD33, PCDH1, JAM2, CDHR3 and IGA3); from the I TDLNS (like CADM1) or from the tumor (like CEACAM6); implying that these molecules could orchestrate the migration of Tregs specifically in each of these tissues. Following the reviewer’s comment:

- we observed that Tconvs from the tumor expressed significant higher gene levels of LFA-1 than Tregs (in the other tissues differences were not significant).

- we also explored which of the total DEG Treg/Tconv shared between two tissues (344 and 42 genes in the Venn Diagram, Figure 6A) were adhesion molecules or molecules related with T cell migration. We observed that Tregs from NI ad I TDLN shared a higher gene expression of CD58, ITGAM, MCAM, CEACAM4, SELPLG and HMMR, which could ensure recirculation among TDNLs (B); and Tregs from I TDLN and tumors shared a higher expression of ICAM1 (C), which could be responsible for the migration between the tissues with presence of tumor cells.

We have mentioned these results in Page 6, L289-297.

For reviewer perusal, we have included below a heatmap with the expression of the above-cited molecules.

LFA-1(ITGAL) Adhesion molecule differentially expressed between Tregs and Tconv in T.	Adhesion molecules differentially expressed between Tregs and Tconv in both LN	ICAM 1 Adhesion molecule differentially expressed between Tregs and Tconv in I-LN and T.
---	---	--

6. Common TCR repertoire may be evidence of pTreg development in the tumor and dLN. Nevertheless, Helios expression in Tregs at the various locations should be mentioned (as in Supp Fig.2a). Although Helios is not a marker exclusively for thymic Tregs, its differential expression indicates the possible level of in situ pTreg differentiation in tumor and dLN.

Answer: We studied by FACS the expression of Helios in the different CD4+ T cell populations. We have now included in supplementary Figure 2D examples of FACS stainings (sup. Figure 2C) and histograms and quantification (sup. Figure 2D), and added results are described in the discussion section page 9 L415.

This panel has now been added as Supplementary figure 2 C and D

7. CD80 was found to be upregulated in Tregs associated with reduced OS and DFS. This was confirmed at the protein level for Tcons and Tregs (Fig. 7I and J). However, since the focus of this

paper is on Treg accumulation within draining lymph nodes invaded by tumor cells, the expression (either RNA or protein) of CD80 in Tregs from NI dLN, I dLN and tumor ought to be presented as well. Could the difference in CD80 expression between HD and cancer patients be detected in PBMC Tregs?

Answer: We agree with the reviewer on the relevance of these comments and we would have done more stainings if possible. However, due to the SARS-CoV19 infection and the measures adopted by French government, our laboratory has been shut down, and it will not be possible for us to do further experiments in a reasonable timeline. Nevertheless, we succeeded to stain TDLN and tumor samples from two patients.

As shown below, the % of CD80+ T cells was slightly higher among Tregs from I-TDLN than among Tconvs (Figure A below). Tumor Tregs expressed the highest frequencies of CD80 compared to tumor Tconv and T cells from I-LNs. Unfortunately, we could not stain in NI-TDLNs, as we do only very occasionally receive NI-LNs, as pathologist reserve them for diagnosis purpose.

Regarding the RNA levels, we observed a similar tendency: CD80 expression is statistically significantly higher in Tregs from tumors than in Tconvs, but the expression level in I-LN is not statistically significantly different (Figure B below). Tregs In NI-LN also showed higher levels of CD80 than Tconvs.

We have now added these results in supplementary figure 6E (Figure S6E), and commented this results in page 7, L314.

Lastly, as the authors deem CD80+ Tregs to be highly stable and activated, it is worth assessing the effect of CD80 blockade on Treg stability and survival. This could be done in vitro with tumor Tregs alone or co-cultured with Tcons.

Answer: Again, we agree with the reviewer on the relevance of this proposed experiment, but due to the SARS-CoV19 infection and the measures adopted by French government, our laboratory has been shut down, and it will not be possible for us to do further experiments in a reasonable time. We alternatively we have enriched the discussion on CD80/Tregs, starting on Page 10 L438.

Minor points:

1. The word 'revealed' is misspelled several times as 'reveled'.

Answer: We thank the reviewer. This have been accordingly modified

2. A brief description of tumor environment (hypoxia?) in breast cancer tissue which Tregs adapt to can be added in discussion.

Answer: We thank the reviewer. This has been added, Page 9 L430.

3. CD79a and TNFRSF13B are conventional B-cell receptors. Can the authors speculate reasons for their possible role/s in rendering Tregs 'fragile', particularly for TNFRSF13B which has been shown to be present on T-cells, contributing to their activation?

Answer: We understand this reviewer's criticism and also of reviewer 2 on our hypothesis of the role of CD79a and TNFRST13B on Tregs. Based on the 2 reviewers comments, we propose not to include our speculative interpretation, but restrict ourselves to the description of the finding, which we find intriguing, but which merits further understanding before elaborating on it. The text has been accordingly modified page 7, L308, L320; and page 8 L364.

Reviewer #2

This study by Gonzalo and colleagues, which is well designed and nicely executed, explores and further characterizes the Treg compartment in ER+ breast tumors and their draining lymph nodes. The authors make some interesting observations that will need further functional validation to confirm their potential clinical impact. On the whole, their observations are rather incremental as (contrary to their claim) flow cytometry based studies have been published before, showing accumulation of CD25^{hi} FoxP3^{hi}(Ki67⁺) effector Tregs in tumor-involved nodes (and -to a lesser extent- even in non-involved nodes, as compared to healthy breast-draining nodes, see van Pul et al J Immunother Cancer 2019 : as well as in tumors, wherein Plitas and colleagues showed functional suppression, PD-1 expression, and also performed transcriptional analyses, comparing Tregs in breast tumors to Tregs in normal breast tissue and PBMC. While the authors make some interesting observations in terms of potentially novel therapeutic targets (CD80 on Tregs), these are mostly observed in tumors rather than lymph nodes and remain to be functionally confirmed as viable targets to reduce Treg numbers or activity.

Answer: We thank the reviewer and we have added the cited reference " Pul et al" in page 2, L70 and in page 8 L348. The work from "Plitas et al" was already cited first in page 2 L80 and several other times along the manuscript, as we agree with the reviewer that it is a key reference in the field.

Specific comments:

1) Why was the actual suppressive activity of Tregs in NI lymph nodes not tested?

Answer: We agree with the reviewer on the relevance of this comparison, and we have previously envisioned doing such experiment. However, the size of the NI TDLN that the pathologists provide is really small, as the *NI LNs are quasi exclusively devoted to the clinical evaluation for diagnosis and staging of the patients*. The samples we have received along the years that took us to do this

work were roughly sufficient to do phenotype and function analysis of unsorted T cell populations, and we could only do the RNAseq from sorted Treg cells from very few patients with occasional bigger NI LNs. Thus, are in the impossibility of performing this experiment in a reasonable timeframe.

2) While Tregs from I TDLN and T show similar in vitro suppressive activity, the proliferative Tconv capacity is reduced in T: how do the authors explain this?

Answer: One explanation to this observation could be that Tregs are more proficient than Tconvs to survive and function in the presence of hypoxia, acidosis and nutrient deprivation (Overacre-Delgoffe, A. E., et al), which can be assumed to be more stringent in the tumor than in the I TDLNs, due to differences in tumor burden.

We have now added this comment in the discussion session, page 8 L359.

They claim their findings from performed suppression assays point to a “stable” Treg phenotype in I TDLN, but since this is an in vitro assay with isolated Tregs, it is questionable that this is representative for their in vivo activity. Also, an absence of IFN-gamma production is no guarantee of active suppression. Please discuss.

Answer: We agree with the reviewer that we cannot directly extrapolate our in vitro results to an in vivo behavior. Nevertheless, in the absence of the tumor in vitro, we can consider that at least the in vivo effects of the tumor on Tregs, did not “imprint” Tregs with a dysfunctional phenotype, what has been observed ex-vivo in other situations as for example Tregs from pancreatic LN of T1D patients (Ferraro A, et al. Diabetes. 2011 Nov;60(11):2903-13); peripheral Tregs from autoimmune vasculitis patients (Wayel H.A., Arthritis Rheum. 2007 Jun;56(6):2080-91), or in different mouse models.

We have taken in to consideration this reviewer’s comment and restrained our conclusion to “ex-vivo” suppression function, Page 4, L173; and Page 8 L355.

Answer: We agree that absence of production of IFN-g is no guarantee of active suppression. However, it has been previously described that a Treg that start producing IFNg (or other pro-inflammatory CKs) can lose the suppressive capacity and acquire effector functions. Indeed, IFNg production identifies dysfunctional Treg cells in patients with multiple sclerosis (Dominguez-Villar, M. Nat. Med. 17, 673–675 (2011), Sumida T, et al .Nat Immunol. 2018 Dec;19(12):1391-1402. (2018), type 1 diabetes (McClymont, S. A. et al. J. Immunol. 186, 3918–3926 (2011); and in glioblastoma (Lowther, D. E. et al.. JCI Insight 1, e85935 (2016).

We understand the way we wrote the phrase was an overstatement, and have now rephrased the conclusion in page 4 L175 integrating this reviewer’s remark.

3) It seems questionable that shared TCR sequences between Tconv and Tregs should be taken as incontrovertible evidence of Treg conversion from Tconv. It could also just mean shared specificity between separate Treg and Tconv clones. Please discuss.

Answer: Because of the size of the TCR repertoire and the high cross-reactivity of the repertoire, it is unlikely that the same TCR sequence would be used to recognize a given MHC:peptide complex. The use of the same TCR in different subsets is suggestive but not demonstrative of a common clonal origin. This has been demonstrated in our recent paper in Nature Communication (Ramirez et al, 2018), where we show that a monoclonal CD4 T cell may become effector, Treg or anergic in the tumor draining LN as soon as 7 days after priming.

4) Based on CCR/CXCR expression patterns, the authors claim a Th1-like profile for Tregs in I TDLN, whereas they have demonstrated an absence of IFNgamma expression. So, what exactly do they

mean by “Th1-like” –doesn’t it rather demonstrate a shared migrational imprinting program with effector (Th1) Tconv, like previously described by the team of Drs. Sallusto and Lanzavecchia? NB: the used gating strategy of CD127-CD25+ here will not just select for Tregs but will also still comprise FoxP3+ activated Th cells.

Answer: Indeed, as this reviewer says, and as explained by us in the manuscript (PAGE 5) line 190, we applied exactly the markers described by the team of Drs. Sallusto and Lanzavecchia. We agree with the reviewers that our results “likely demonstrate a shared migrational imprinting program with effector (Th1) Tconv”.

We have now added this commentary in page 5, L212.

Nevertheless, Tregs, unless they are “unstable” do not produce IFN-g (Figure 3B), yet, they can express CCR/CXCR molecules and also T-bet (which we have now analyzed by FACS; see response to question 3 from reviewer 1), reminiscent of a TH1-like phenotype. We have now added these results as Supplementary figure 4C and D.

For the last part of this reviewer’s remark, we had the same concern on the gating strategy of CD127-CD25+ cells. That is why in the manuscript we had included the supplementary Figure 4 C and D and the text below in page 5 L203.

“The permeabilization step required for FOXP3 staining is not compatible with labeling of chemokine receptors, except for CXCR3 and CCR4. Consequently, we further studied the expression of these two receptors in the CD4+ T cell subsets as described in **Figure 2A**. We observed that the frequency of CXCR3+ cells was significantly increased among Eff Tregs (57,9% vs 67,5%, $P < 0.01$), FOXP3+ non-Tregs (38,9% vs 47%, $P < 0.01$) and memory Tconvs (42,3% vs 53,5%, $P < 0.01$) from the I compared to NI TDLNs (**Figure S4A**) and we observed similar frequencies of CCR4+ CD4+ T cell subpopulations in I vs NI TDLNs (**Figure S4B**). »

Moreover, the newly added supplementary figure 4 C-D and the lines discussing these results in in the page 5, L209, should allay this reviewer’s concern.

5) Their use of CD80 and CD79a transcript levels “normalized “ to FoxP3 (by using a ratio) seems odd...how does this guarantee actual co-expression? This would require scRNAseq?

Answer: We agree that this is not a direct approach, but it has been previously successfully used in literature as a first step to query data by a bioinformatic unbiased approach (as in Plitas et al Immunity (2016); Freeman ZT et al, J Clin Invest, 2020), which is then validated at the protein level

(as in these papers and in ours). Thus, the co-expression in our case has been validated at the protein level (in figure 7I CD80 expression is shown on Tregs, which are gated as FOXP3+ cells). scRNAseq would effectively be another way of showing co-expression.

Per reviewer perusal, please find below some examples of FACS dot plots illustrating the co-expression of FOXP3 and CD80 in tumor Tregs (plots are gated in CD4+ T cells).

6) Just an association with improved OS for e.g. Treg/CD79a, does not necessarily identify them as fragile Tregs: please rephrase and avoid such overstatements without functional/phenotypic confirmation.

Answer: We understand this reviewer criticism and also of reviewer 1 on our hypothesis of the role of CD79a and TNFRST13B on Tregs. Based on the 2 reviewers comments, we propose not to include our speculative interpretation, but restrict ourselves to the description of the finding, which we find intriguing, but which merits further understanding before elaborating on it. The text has been accordingly modified page 7, L320 and page 8 L364.

7) Was there a relationship between Tregs and the activation state of Tconv, including CD8+ effector T cells in NI TDLN, I TDLN and T?

Answer: We did not observe any significant correlation between the frequency of Tregs and the frequency or activation state of CD8 T cells, NK or B cells, probably due to the inter-patient heterogeneity and the relatively small number of samples. However, in the I TDLNs and the Tumors we found two positive correlations of the % of Tregs vs IFN+Tconvs and the % of Tregs vs PD1+Tconvs respectively.

Per reviewer perusal, please find below the two examples. Since we did not observe a clear

pattern of correlation in the nodes and tumor we consider not including the plots in this manuscript.

8) The observation of the presence of CD80+ Tregs in T is interesting: what about in NI TDLN and I TDLN?

Answer: Please see answer to question 7 from reviewer 1

A possible relationship to OS warrants discussion of a possible underlying hypothesis. The authors mention CTLA4-mediated trans-endocytosis but don't refer to the paper in question: it certainly deserves more discussion in relation to their identification of CD80+ Tregs; in this regard, how certain can they be of membrane vs. i.c. expression of CD80, as they apparently performed staining both before and after permeabilisation?

Answer: We thanks the reviewer for the observation about the missed reference. We added it in page 7, L318 and page 10, L440 (reference : Ovcinnikovs V. et all, Science Immunology, 2019).

Regarding the CD80 staining, due to the limited number of CD80+ cells in our samples, and based in the very recent results published in the preprinted paper <https://doi.org/10.1101/706275> (where authors describe that CD80 marks recently activated T cells in circulation and show that the double staining performs better for this marker) we decided to do the CD80 staining in two steps: 1) live/dead staining and surface staining with CD45, CD3, CD4, CD80, CCR8; and then, 2) we fixed and permeabilized the samples and did an intracellular staining, with FOXP3 and CD80. So, as the reviewer points out, in this experiment we are not able to discriminate membrane vs intracellular CD80 protein expression. Of note, the presence of CD80 protein in the membrane or IC, will not help to distinguish whether CD80 was synthesized by the Treg or acquired from a DC membrane, as Tregs can internalize the captured ligands (Ovcinnikovs V. et all, Science Immunology, 2019). However, detection of the CD80 mRNA by RNAseq can be taken as an indicative of production of CD80 by the Treg.

Following this reviewer's remarks, we have now further discussed this issue in the results section, starting in the Page 7, L316; and we have enriched the discussion, starting on Page 10 L438.

9) Finally, was there a relationship between Treg frequencies/activation state and 1) distance of TDLN to tumor, 2) size of the mets in I TDLN, 3) HR and Her-2/neu status: it is a pity the study only included patients with HR+ tumors. Certainly, in terms of impact for possible immunotherapy applications, the inclusion of triple negative tumors and their TDLN would have been highly relevant.

Answers:

-We do not have access of the information of the TDLN-tumor distance, neither of the size of the metastasis, as the pathologists do not record this data.

- For the I TDLNs we do evaluate by FACS the percentage of EPCAM+ CD45+ cells, which is enriched in tumor cells. However, as we only receive pieces of TDLNs, it is not possible for as to have an accurate measurement of the size of the metastasis.

Concerning patients with HER2+ or TN tumors, we could only analyze few of them. Our preliminary results obtained from HER2 and triple-negative breast cancers suggest that our findings might reflect a phenomenon shared by other types of breast cancer. However, the number of patients is limited, so we could not apply statistical tests.

We show these results below for reviewer perusal, but we consider them too preliminary to be included in this manuscript.

Clinical and Pathological data of the Breast cancer Patients					
Characteristics	Patients	Value		Patients	Value
Age (years)	N= 4		Histology		
Lymph node status			Tumor size (cm)		
N1 (1-3)	2	50%	T1 (≤ 2)	3	75%
N3 (>9)	2	50%	T2 (2-5)	1	25%
Stage			Hormone receptor status		
II	2	50%	Triple negative	3	75%
III	2	50%	ErbB-2 receptor tyrosine kinase 2 (HER2)	1	25%

Global distribution of T cell populations in NI and I TDLNs from HER2 and triple negative breast cancers. Axillary TDLNs cells were stained for CD3, CD4, CD45RA, FOXP3, CD27, PD-1 and CTLA-4. A) Representative flow cytometric analysis (left panel) of cells expressing CD45RA and FOXP3 among CD4+ T cells, shown for one HER2+ breast cancer patient. Quantification (right panel) of the different selected population in 3 TN and one HER2+ patient. B) Quantification of PD-1 (left panel) and CTLA-4 (right panel) among different CD4+ T cell populations in a TN breast cancer patient. It can be appreciated that Tregs in the I TDLN express higher levels of PD1 and CTLA-4 than in the NI TDLN (as for luminal breast cancer patients).

Reviewer #3 (Remarks to the Author):

It's true that detailed molecular and functional characterization of T cell subsets in draining lymph nodes of breast cancer and other cancer types is limited. Gonzalo et al. provides a relatively comprehensive profile of CD4 T cell subsets, mostly focusing on CD4 conventional and regulatory T cells (Treg), in matched tissues of positive and negative draining lymph nodes and lumina breast tumors. This study combines information of cellular composition, protein marker expression, bulk RNA transcription and TCR repertoire analysis. The strength of the study is comparative analysis on matched tissues of negative lymph nodes, positive lymph nodes and tumors in the same individual, which is definitely welcome and would provide useful information to the community. The authors find Treg is the only cell subset with higher proportion in invaded lymph nodes than

the non-invaded ones, among all cell subsets they investigate. These Tregs in invaded lymph nodes express higher level of immune checkpoint molecules, reserve activating suppressive function and follow a similar polarization pattern as conventional T cells (Tcon). TCR repertoire result supports a proportion of conversion from Tcon to Treg. Bulk RNA sequencing identifies common Treg transcription signatures and tissue specific signatures, in which CD80 expression correlates with poor survival from TCGA samples. The whole study is well designed, and multiple state-of-art technologies are used. Most conclusions can be supported by the data, though some claims lack solid evidences. Overall, the story starts from an attractive finding (accumulative Treg cells in invaded lymph node) and provides rich information, however, the whole story seems to lack some striking highlights. Some important points mentioned are either consistent findings or lack further evidences. I will elaborate some points as follows.

1-L109-L111: “almost each” is an over statement. Treg proportions are lower in 6 out of 14 samples, and N=14 should be added in Fig. 1F.

Answer: We have specified that the number of patients for which the proportion of Tregs is higher in the I vs the NI TDLNs is 10 out of 14; and N=14 was added in Fig. 1F.

The sentence was modified, Page3, L109.

For reviewer perusal we have included below the raw data.

% Tregs/CD45+ cells		
Treg NI TDLN	Treg I TDLN	Treg Tumor
0.5702161	1.486043	1.607462
1.66824	2.990778	14.0072
2.577861	5.013492	17.36295
0.3572258	0.3321375	0.9267764
0.4728367	1.216951	3.991803
0.2138048	0.5850101	0.2930708
0.7694547	0.7592349	2.966741
2.170992	2.67228	1.117652
0.9519821	1.325033	2.087795
0.2383123	0.2215988	0.1887513
0.4780735	0.412267	0.7954213
0.7248576	0.7397285	5.505618
0.1785036	1.048015	0.3068965
0.6341217	0.6784999	0.7266529

2. L435 I cannot find supplementary table 8

Answer: We thank the reviewer. Yes, this table was missing; we have now put it back

3. L219 “that” should be “than”

Answer: We thank the reviewer. This have been accordingly modified. Page 6, L229

4. There are some problems with the TCR repertoire analysis. Firstly, the authors miss plenty of important details in the method section, which would leave obstacles for the reproducibility of the experiment. No information is provided about the quantity of input RNA and the correspondent cell number, for the RT reaction. Are all the sorted cells used? Three PCR steps are used for the

TRB library construction, but no input DNA quantity is mentioned for each step. The key is whether all the sorted cells listed in supplementary table 3 are processed to sequencing.

Answer: Indeed, this is an important point. The RNA was extracted from all the sorted cells. After purification, all the RNA was used for the reverse transcription step. 17 cycles were used for the first PCR step to keep it in the exponential phase allowing > 10000-fold amplification and an unbiased representation of all the TCR β transcripts. After beads-based purification, 1/160th of the purified product was used as template for the second PCR allowing a good representation of the initial template. We attempted to adapt the number of reads per sample according to the initial number of cells.

We have now added this information in the material and method section. Page 11-12 beginning in L546

If yes, it seems the produced sequencing reads are not enough to cover all the cells (especially for the conventional CD4), which will underestimate the number of clones. The authors haven't used any normalization method, which would make the number of clones being not representative of the number of sorted cells. There are three major conclusions in this section. 1) clonal expansion: tumor>invaded lymph node>non-invaded lymph node; 2)trafficking between tumor and lymph node; 3) conversion between Tregs and conventional T cells; For the first two points, the authors use the top 100 clones to perform the comparison, which won't be influenced too much by the problem I mentioned above.

Answer: We agree with the reviewer that the number of reads might have been insufficient to exhaust the whole TCR β repertoire of the conventional T cell samples. However, we studied only memory T cells, which encompass clonal expansions. Because the number of clonotypes was much higher in the I and NI LN than in the tumor samples and the average number of reads per clonotypes was most often >10 (and always >6.7), we don't think that we have missed important clonal expansions.

On the other hand, normalization methods on RNA extraction on variable number of highly different proportion of clonal cells, each one containing 10-50 copies of RNA are complex to set up without performing complex bio-informatic modeling. In our opinion, the best method would be to study similar number of cells. However, in this work, as some samples contained very few cells, we would have lost a lot of information if the number of studied cells would have been aligned to the smallest samples. For this reason, we chose to only study the 100 most abundant clones as acknowledged by the reviewer. We appreciate the limits of this strategy.

However, for point 3, the authors are not stringent to use all the clones in the data to calculate the shared percentage between Tregs and conventional T cells. Further, when calculating the percentage of shared clones, the authors should indicate whether the frequencies of clones are taken into consideration.

Answer: We thank the reviewer for the observation. We have consequently specified in the material and methods Page 12, L550 that the percentage of shared clones was calculated as follow: number of shared clones/ total number of Tregs clones x 100, for each tissue.

5. L235, Could the authors provide some biological significance for the inter-tissue sharing?

Answer: In the discussion section, on top of the contribution of TDLNs as source of tumor-specific T cells that might be modulated by immunotherapies, we have added two points of discussion: (Page 9 L408):

As suggested by Zemmour et al (ref), Tregs with the same TCR have similar transcriptional traits, implying that Tregs with shared antigenic specificity from tumors and TDLNs may also share an early imprinted specific program that persists after priming and drive them to anatomical locations with common environmental cues. Furthermore, inter-tissue sharing of Treg clones could also be indicative of a loco-regional suppression mechanism.

6. L381-384, when the authors discuss the sharing of TCR clones between tumor and TDLN they are preferable to mention previous reports (such as reference 83), though it's interesting to see similar result in sorted conventional and regulatory T cells.

Answer: we agree with the reviewer, but we do understand if the reviewer wants us to do a specific action following his comment.

7. CD80:FOXP3 ratio are identified to be associated with worse survival from the TCGA breast cancer data. As one of the DEGs between regulatory and conventional T cells, is CD80 a “common signature” or adapted to certain tissues?

Answer: please see answer to reviewer 1, question 7.

8. CD80 is conventionally thought to be expressed in dendritic cells and other antigen presenting cells. When the authors discuss CD80 expressing regulatory T cells in tumor and use FACS to validate it, they shouldn't neglect previous reports about acquisition of CD80 to T cells upon activation (DOI: <https://doi.org/10.4049/jimmunol.169.11.6162>, <https://doi.org/10.1038/cmi.2011.62>). At least they are preferred to discuss why they think CD80 are expressed by T cells rather than being acquired from antigen presenting cells.

Answer: please see answer to reviewer 2, question 7.

We thank the reviewer for this comment, the cited references have now been commented and added, in the results section, starting in the Page 7, L317; and we have enriched the discussion, starting on Page 10 L438.

9. Inconsistent use of “tables” and supplementary tables”

Answer: We thank the reviewer. This have been accordingly modified

REVIEWERS' COMMENTS:

Reviewer #1 (Remarks to the Author):

The authors have properly responded to my comments.

Reviewer #2 (Remarks to the Author):

Overall Gonzalo and colleagues provide some interesting observations, but in terms of novel insights and influencing the way of thinking in the field, their contribution remains limited. It is already known and has been described that tumor involvement in breast cancer-draining lymph nodes is accompanied by increased rates of activated Tregs. There are however some intriguing additional findings (like shared TCR repertoire of Tregs in the tumor and its draining lymph nodes, a shared chemokine receptor profile with Th1-like cells and CD80 expression related to survival). Importantly, the (functional) comparison of NI vs I LN is limited and thereby also reduces the impact of the findings. Finally, there are still some overstatements/conclusions that in my opinion are not fully backed by the data.

I have remaining comments pertaining to my original review and the authors' subsequent rebuttal:

- 1) Issue 3, the author's rebuttal to this point should be integrated into the actual paper with references provided
- 2) Issue 4: I maintain that a shared chemokine receptor profile with Th1 cells does not justify referring to the Treg phenotype as Th1-like, even though there may be Tbet expression; it suggests a functionality that is not demonstrated. Please delete statements like "Th1 biased polarization of T cells"
- 3) Issue 5: Please discuss previous papers on this bioinformatics approach in the paper and also include the FoxP3/CD80 FACS plots in the paper: they provide important information and validation
- 4) there is still a reference on p3 to CD79a/TNFR13B+ Tregs as "fragile"; please remove.

Reviewer #3 (Remarks to the Author):

The authors have satisfactorily addressed most of my concerns. Though I still think the number of sharing percentages between Treg and Tcon are not accurate, I won't further argue it since the number should have been underestimated and wouldn't impact the point. For point 6 I raised, the authors should cite reference 83 when talking about T cell trafficking between tumor and TDLN, because this reference has already described it. A typo is to be corrected in L318, "of both"--"or both".

Answer reviewer comments 2

Reviewer 2

1. Issue 3, the author's rebuttal to this point should be integrated into the actual paper with references provided

We have integrated our response and accordingly modified the text starting in line 348 as follows:

Although the use of the same TCR in different subsets is suggestive but not demonstrative of a common clonal origin, it has been recently shown in a mouse model³⁷ that monoclonal CD4⁺ T cell may become effector, anergic, or Treg in the TDLNs as soon as 7 days after priming.

2. Issue 4: I maintain that a shared chemokine receptor profile with Th1 cells does not justify referring to the Treg phenotype as Th1-like, even though there may be Tbet expression; it suggests a functionality that is not demonstrated. Please delete statements like "Th1 biased polarization of T cells"

We agree with the reviewer suggestion and we deleted “polarization” in the lines 166,174, 178, 190, 324.

3. Issue 5: Please discuss previous papers on this bioinformatics approach in the paper and also include the FoxP3/CD80 FACS plots in the paper: they provide important information and validation

The FOXP3/CD80 representative plot is now included in the supplementary figure 6-G. The bioinformatics strategy is detailed in the result section, and we have further developed the bioinformatics approach in material and methods, under the Survival analysis heading (page 11), as follows:

“As overall survival (OS) and disease free-survival (DFS) of FOXP3 mRNA expression level per se is not a good read out of the well known contribution of Tregs to tumor escape, we used a bioinformatics strategy, as proposed by Plitas et al 2. Briefly, first, for the survival analysis we selected from the list of DEGs obtained from Figure 6A, only those genes that positively correlated ($R > 0.5$ and counts > 20 RPM) with FOXP3 reads per million of mapped reads (RPM). Then, to evaluate the gene ratios, DEG RPMs were normalized by FOXP3 RPM and the obtained DEGs:FOXP3 proportions were used to segregate the breast cancer patients in two groups (low and high) based on the median expression level. Statistical significance of the curves (OS and DFS) stratified by the above-mentioned groups was determined using a Log-rank test (R package survival).”

4. There is still a reference on p3 to CD79a/TNFR13B+ Tregs as "fragile"; please remove.

The word fragile was now removed.

Reviewer 3

1. For point 6 I raised, the authors should cite reference 83 when talking about T cell trafficking between tumor and TDLN, because this reference has already described it.

Reference 83 (now 35) from the previous version (Wang et al), was included in the reviewed manuscript in the line 448 (now 336), when we discuss the trafficking between tumor and TDLN.

We have now included the following sentence when we describe the analysis (Line 196):

“The TCR repertoire has been used to explore the clonal diversity and trafficking patterns among TDLNs and tumor of total T cells³⁵. To go deeper in CD4+ T cells subsets, we performed high-throughput...”

2. A typo is to be corrected in L318, "of both"--"or both".

This has been corrected.